# Towards Completeness in Causal Discovery from Soft Interventions with Known Targets

**Zihan Zhou** [1]  **Murat Kocaoglu** [1]

## Abstract

We study causal discovery from soft interventions in the presence of latent confounding. Beyond within-environment conditional independences, soft interventions induce cross-environment invariances that can be encoded using an augmented graph with intervention indicator nodes ($\mathcal{I}$-AUG). Taking its maximal ancestral graph (MAG) yields the $\mathcal{I}$-MAG, which characterizes the interventional Markov equivalence class. Building on this framework, we show that the FCI-inspired learner ($\mathcal{I}$-FCI) by Kocaoglu et al. (2019) is sound but not complete: it may output circle endpoints that are nevertheless compelled by the interventional equivalence class. To exploit intervention-node semantics, we propose two complementary methods. First, we introduce an enumeration-based completion procedure that is sound and theoretically complete, but whose worst-case cost depends on the number of MAGs compatible with the partial graph learned by $\mathcal{I}$-FCI. Second, we derive a set of additional local orientation rules that provably tighten $\mathcal{I}$-FCI without increasing asymptotic complexity. Both methods refine prior outputs in the controlled soft-intervention setting with latent variables.

## 1. Introduction

Causal discovery aims to infer the underlying causal structure of a system from data—typically a mixture of purely observational measurements and, when available, data collected under interventions (Pearl, 2009). Recovering causal relationships is central to reliable prediction under distribution shift and to principled decision making, since causal models support queries that correlation alone cannot answer (e.g., what will happen if we change $X$?). This has made causal discovery increasingly relevant across domains such as drug development and precision medicine (Hernán & Robins, 2016; Sanchez et al., 2022), where interventions correspond to treatments; genomics (Meinshausen et al., 2016; Belyaeva et al., 2021), where perturbation experiments probe regulatory mechanisms; microservice systems (Ikram et al., 2022; 2025), where system failure can be modeled as an intervention to the system; and policy analysis (Athey & Imbens, 2017; Abadie et al., 2010), where causal conclusions guide high-stakes interventions. At the same time, modern datasets often arise from heterogeneous environments, i.e. different experimental conditions, deployments, or regimes, creating both new opportunities and new challenges for causal structure learning. A core question is how to systematically translate such multi-environment data into additional, identifiable causal information.

A common formal framework for reasoning about causal systems is the structural causal model (SCM), in which a directed acyclic graph (DAG) encodes how each variable is generated from its direct causes and an exogenous noise term. In this representation, nodes correspond to system variables and directed edges represent direct causal influence. Interventions can be modeled as modifications of the data-generating mechanisms. A hard intervention sets a variable's value while a soft intervention perturbs its conditional distribution while leaving the rest of the system intact. Graphical causal models provide a compact language for expressing both qualitative causal relationships and the invariances they imply. In realistic scientific and engineering settings, however, not all relevant variables are observed. Latent confounders can induce spurious dependencies among observables (Greenland et al., 1999; Ali et al., 2009), complicating causal discovery and motivating mixed-graph representations over the observed variables.

A classical approach to causal discovery leverages conditional independence (CI) constraints. Under the Markov and faithfulness assumptions, CI relations in the observed distribution correspond to graphical separation (Verma & Pearl, 1992) properties in the underlying causal graph. Constraint-based algorithms exploit this correspondence by performing CI tests to estimate the graph's adjacencies and some

---

[1]School of Computer Science, Johns Hopkins University, Baltimore, United States. Correspondence to: Zihan Zhou <zzhou150@jh.edu>.

*Proceedings of the 43rd International Conference on Machine Learning*, Seoul, South Korea. PMLR 306, 2026. Copyright 2026 by the author(s).

edge orientations. In the fully observed setting, the PC algorithm and its variants (Spirtes et al., 2001) recover a completed partially directed acyclic graph (CPDAG) that represents a set of equivalent DAGs called the Markov equivalence class (MEC). With latent confounding, algorithms such as FCI (Zhang, 2008) output a partial ancestral graph (PAG), which encodes the MEC of maximal ancestral graphs (MAGs) consistent with the observed CI constraints. Consequently, purely observational CI information typically determines only an MEC rather than a unique causal graph, leaving edge marks undetermined whenever they are not compelled by the invariance constraints. To further recover the causal structure, one needs either interventional data or additional parametric assumptions on the SCMs.

Interventional data can break this observational ambiguity by introducing cross-domain invariances. In the fully observed setting, the impact of interventions is well understood. In particular, for families of interventions one can define an interventional Markov equivalence relation that is finer than the observational one, together with a canonical graph representation that generalizes the CPDAG. The interventional essential graph of Hauser & Bühlmann (2012) provides a complete characterization and enables consistent learning of all edge directions. When latent variables are present, interventional constraints must be combined with mixed-graph reasoning, and the appropriate aim is an interventional equivalence class. Kocaoglu et al. (2019) address this challenge for controlled soft interventions, deriving testable cross-domain invariances from soft do-calculus (Correa & Bareinboim, 2020) and encoding them through an augmented graph construction that introduces intervention indicator nodes connected to the perturbed targets. The resulting object can be summarized as an augmented MAG called $\mathcal{I}$-MAG over observed variables and intervention indicator $F$ nodes, yielding a characterization of the interventional Markov equivalence class ($\mathcal{I}$-MEC), and a constraint-based learning algorithm inspired by FCI, later referred to as $\mathcal{I}$-FCI by Li et al. (2023).

In Figure 1, we show an example of how to construct the augmented MAG given the causal graph and intervention targets. The output of $\mathcal{I}$-FCI is shown in Figure 1d. Despite this progress, the learning procedure of $\mathcal{I}$-FCI is sound but can return augmented PAGs containing circle marks that are in fact identifiable from the same set of observational and interventional invariances. We demonstrate this gap via a simple counterexample in Figure 1, where the learned output leaves an ambiguous edge even though all compatible $\mathcal{I}$-MAGs agree on its orientation. The key phenomenon is that adjacency patterns involving intervention nodes constrain the existence and structure of inducing paths through the intervention targets. In particular, since an intervention node $F$ has outgoing edges only to variables whose mechanisms differ across the compared regimes, any inducing

path from $F$ to a non-target variable must first pass through one of those intervened targets. These constraints reflect the semantics of the augmentation and are not fully propagated by the generic orientation closure used in prior works.

Motivated by this observation, we develop two approaches to extend $\mathcal{I}$-FCI : an enumeration-based, theoretically complete completion procedure using MAG listing and an efficient local orientation rule based learning algorithm. Both exploit intervention-node semantics to resolve the ambiguities. Our contributions can be summarized as follows:

- We show that the FCI-inspired orientation procedure of $\mathcal{I}$-FCI is *not complete*: there exist instances where it outputs circle endpoints that are nevertheless compelled by the same controlled soft-intervention semantics, and we explain the structural reason.

- Building on these insights, we propose an *efficient refinement* of the baseline algorithm by adding three local orientation rules that are sound under the assumptions of Kocaoglu et al. (2019) and tighten the learned graph without increasing asymptotic complexity.

- We propose an enumeration-based completion procedure for multi-domain interventional data. We treat the post–$\mathcal{I}$-FCI output as a PAG with local BK, enumerate consistent MAG completions via MAG listing, and filter via an $\mathcal{I}$-MAG realizability oracle. This yields a sound and theoretically complete procedure, though it can be expensive when many MAG completions exist.

- We evaluate the fast rule-based refinement and enumeration-based completion procedure on synthetic data, showing that our methods can recover more edge marks than $\mathcal{I}$-FCI .

## 2. Related Works

**Causal discovery.** Broadly, causal structure learning methods fall into three categories: constraint-based approaches which use CI tests, score-based approaches which search for a graph maximizing a goodness-of-fit score, and optimizations which turn DAG learning into a differentiable program. Here we focus on the constraint based methods since we are also using CI statements for the discovery task. The classic constraint-based pipeline is developed in (Spirtes et al., 2001) and includes PC-style methods for causally sufficient systems and FCI-style methods for settings with latent confounding and selection bias (Zhang, 2008). Importantly, many widely used DAG-learning methods are primarily formulated for causally sufficient DAGs. By contrast, explicitly accommodating latent confounders generally requires mixed-graph formalisms and algorithms such as FCI and its variants.

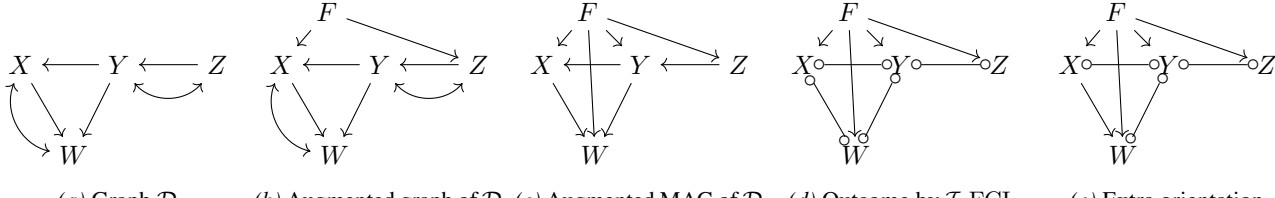

*(a)* Graph $\mathcal{D}$     *(b)* Augmented graph of $\mathcal{D}$     *(c)* Augmented MAG of $\mathcal{D}$     *(d)* Outcome by $\mathcal{I}$-FCI     *(e)* Extra orientation

*Figure 1.* An example showing the construction of augmented graphs and augmented MAGs from a causal graph, and that the learning algorithm in Kocaoglu et al. (2019) is not complete. Figure 1a is the original causal graph $\mathcal{D}$, with intervention targets $\mathcal{I} = \{\emptyset, \{X, Z\}\}$. Figure 1b is the augmented graph of $\mathcal{D}$. It adds an additional $F$ node and points it to $X, Z$ since it is the symmetric difference of the two intervention targets. Figure 1c is the augmented MAG of $\mathcal{D}$. It is obtained by taking the MAG of the augmented graph of $\mathcal{D}$. Figure 1d shows the learning outcome of $\mathcal{I}$-FCI in Kocaoglu et al. (2019). Figure 1e shows that $X \rightarrow W$ can be identified. $W$ is not a target but adjacent to $F$. This could happen only when there is an inducing path from $F$ to $W$ via a target. If $Z$ is the target, this implies an inducing path from $Z$ to $W$ which will make $Z$ and $W$ adjacent in the augmented MAG. Therefore, such inducing path can only go through $X$ which allows us to orient $X \rightarrow W$. This process only requires a verification based on local structures and thus can be converted into a local orientation rule. Similarly, we can infer that there is an inducing path connecting $F$ and $Y$ through $X$ or $Z$. Since either is possible, we cannot orient any more edge.

**Markov equivalence and interventional equivalence.** With observational data, CI information typically identifies only a Markov equivalence class (MEC) rather than a unique graph. In the latent-free DAG case this is represented by a CPDAG (Verma & Pearl, 1992). With latent variables, Richardson & Spirtes (2002)'s ancestral graph framework motivates MAGs and their equivalence classes represented by PAGs, and Markov equivalence for MAGs is characterized by Ali et al. (2009). Notably, Zhang (2008) provided orientation rules yielding completeness as FCI algorithm. For fully observed DAGs under known interventions, Hauser & Bühlmann (2012) and Yang et al. (2018) characterize interventional MEC and introduce the interventional essential graph. With latent confounding, Kocaoglu et al. (2019) characterize an $\mathcal{I}$-MEC for controlled soft interventions via an augmented MAG construction. Jaber et al. (2020) extend this framework to unknown targets setting and proposes $\psi$-MEC. Zhou et al. (2025) develop a complementary characterization for hard interventions. These graphical representations for MEC tell us what is fundamentally learnable by any learning algorithm.

**Learning from background knowledge.** A separate but closely related literature studies how background knowledge (BK) can be incorporated into causal discovery to further restrict the equivalence class and compel additional orientations. Relevant to our setting, Wang et al. (2022) study local BK and provide sound-and-complete rule systems for orienting PAGs with latent variables given such BK. Based on the sound and complete rules with local BK, Wang et al. (2024) and Wang et al. (2025) establish the constraints for valid local orientations and develop an MAG listing algorithm to avoid brute force search. Meanwhile, intervention indicator nodes are used in the literature of causality (Eberhardt & Scheines, 2007; Dawid, 2002; Hauser & Bühlmann, 2012). Conceptually, the intervention indicator augmentation used

in interventional causal discovery can be viewed as introducing exogenous context variables with known structural semantics, which functions as a form of BK that can be propagated beyond generic PAG orientation closure. However, this semantics is richer than ordinary local BK: local BK typically fixes edge marks incident to specified nodes, whereas known intervention targets also restrict how an intervention node can become adjacent to variables that are not directly intervened on. When targets are unknown, the main available structural information is that intervention indicators are exogenous source nodes. When targets are known, their known outgoing neighborhoods impose additional constraints on the augmented graph. This connection motivates developing additional orientation rules that explicitly exploit such semantics as we will show.

## 3. Preliminaries

In this section, we briefly describe related background knowledge and notations used in this paper. Throughout this paper, we use bold letters for sets.

**Causal Bayesian Networks (CBNs).** Let $\mathbf{V} = \{V_1, \ldots, V_p\}$ be a set of observed variables and let $\mathcal{D} = (\mathbf{V}, E)$ be a DAG. For a node $V_i \in \mathbf{V}$, let $\mathrm{Pa}_{\mathcal{D}}(V_i)$ denote its parents in $\mathcal{D}$. A directed edge $X \rightarrow Y$ means $X$ is $Y$'s direct cause. A causal Bayesian network associated with $\mathcal{D}$ specifies an observational joint distribution

$$P(\mathbf{v}) = \prod_{i=1}^{p} P(v_i \mid \mathrm{Pa}_{\mathcal{D}}(V_i)), \qquad (1)$$

and implies CI constraints among $\mathbf{V}$ characterized by the $d$-separation criterion (Verma & Pearl, 1992; Pearl, 2009).

**Soft interventions.** We consider a collection of (possibly empty) intervention targets $\mathcal{I} \subseteq 2^{\mathbf{V}}$. For each $I \in \mathcal{I}$, a *soft*

*intervention* modifies the conditional mechanisms of the targets while leaving the remaining conditionals unchanged:

$$P_I(\mathbf{v}) \; = \; \prod_{V_i \in I} P_I(v_i \mid \mathrm{Pa}_{\mathcal{D}}(V_i)) \prod_{V_i \notin I} P(v_i \mid \mathrm{Pa}_{\mathcal{D}}(V_i)) \,.$$
(2)

In later sections we adopt the standard *controlled* soft-intervention assumption used in prior work[1].

**Mixed graphs and latent confounding.** In many applications, some relevant variables are unobserved. We model this via an underlying causal DAG $\mathcal{D} = (\mathbf{V} \cup \mathbf{L}, E)$, where $\mathbf{L}$ is a set of latent variables. The observed joint distribution is obtained by marginalizing out $\mathbf{L}$:

$$P(\mathbf{v}) \; = \; \sum_{\mathbf{l}} \prod_{T \in \mathbf{V} \cup \mathbf{L}} P(t \mid \mathrm{Pa}_{\mathcal{D}}(T)) \,.$$
(3)

Latent common causes can induce spurious dependencies among observables and are naturally represented by *acyclic directed mixed graphs* (ADMGs), which allow both directed edges ($\rightarrow$) and bidirected edges ($\leftrightarrow$). A bidirected edge $X \leftrightarrow Y$ informally indicates the presence of an unobserved common cause of $X$ and $Y$.

**Ancestral and maximal ancestral graphs.** A mixed graph is *ancestral* if it contains no directed cycle and no "almost directed cycle" (i.e., one cannot follow a directed path from $X$ to $Y$ and also have an arrowhead into $X$ from $Y$). A path between two distinct vertices is an *inducing path* relative to a set $\mathbf{Z}$ if every non-endpoint vertex outside $\mathbf{Z}$ is a collider on the path and every collider is an ancestor of at least one endpoint. A path $\pi = \langle X, Q_1, \ldots, Q_p, V, Y \rangle$ ($p \geq 1$) in an ancestral graph is a *discriminating path for $V$* if $X$ is not adjacent to $Y$, $V$ is adjacent to $Y$ on $\pi$, and every $Q_i$ is a collider on $\pi$ and a parent of $Y$. A mixed graph is *maximal* if every pair of non-adjacent vertices can be $m$-separated by some conditioning set; equivalently, there is no inducing path between any pair of non-adjacent vertices. A *maximal ancestral graph (MAG)* is a mixed graph that is both maximal and ancestral (Richardson & Spirtes, 2002; Ali et al., 2009). MAGs encode CI relations among observables via $m$-separation. A *partial ancestral graph (PAG)* represents a Markov equivalence class of MAGs, using endpoint marks as arrowhead, arrowtail or circle mark ($\circ$), where $\circ$ denotes an undetermined endpoint (Zhang, 2008). We allow ADMGs to contain bows ($X \leftarrow Y$ and $X \leftrightarrow Y$ simultaneously). MAGs remain ancestral and thus bow-free.

**Notation.** For any causal graph $\mathcal{G}$ and node $X \in \mathbf{V}[\mathcal{G}]$, we write: $\mathrm{Pa}_{\mathcal{G}}(X)$ (parents), $\mathrm{Ch}_{\mathcal{G}}(X)$ (children), $\mathrm{Adj}_{\mathcal{G}}(X)$ (adjacent vertices), $\mathrm{An}_{\mathcal{G}}(X)$ (ancestors), and $\mathrm{De}_{\mathcal{G}}(X)$ (descendants). For sets, $\mathrm{An}_{\mathcal{G}}(\mathbf{S}) = \bigcup_{X \in \mathbf{S}} \mathrm{An}_{\mathcal{G}}(X)$ and similarly for $\mathrm{De}_{\mathcal{G}}(\mathbf{S})$. The vertices-induced subgraph on $\mathbf{S}$ is

$\mathcal{G}[\mathbf{S}]$. A triple $\langle X, Y, Z \rangle$ is *unshielded* if $X{-}Y$ and $Y{-}Z$ are edges but $X$ and $Z$ are non-adjacent. It is an *unshielded collider* if both edges have arrowheads into $Y$. For intervention indexing, we write $\mathcal{I}$ for the set of available intervention targets, $P_I$ for the distribution in regime $I \in \mathcal{I}$, and $I \Delta J$ for the symmetric difference between two target sets $I$ and $J$. Graph separation statements will be written as $X \perp_{\mathcal{G}} Y \mid \mathbf{Z}$, where the separation criterion is $d$-separation for DAGs and $m$-separation for mixed graphs. $\mathcal{G}_{\overline{X}}/\mathcal{G}_{\underline{X}}$ is the graph obtained by removing all the edges into/out of $X$ from $\mathcal{G}$. For $\mathcal{G}_{\overline{X}, Y(Z)}$, $Y(Z)$ is the subset of $Y$ that are not ancestors of $Z$ in the graph $\mathcal{G}_{\overline{X}}$. $[\mathcal{G}]$ denotes the MEC of $\mathcal{G}$. In a PAG, a circle mark in an edge $X \circ{\rightarrow} Y$ can be either an arrowtail or an arrowhead which is not determined. A star mark in an edge $X {*}{\rightarrow} Y$ is used as a wildcard which can be a circle, arrowhead, or arrowtail. We assume that there is no selection bias. If a node has no incoming arrowheads incident to it, we call it a source node. We omit the subscripts when the graph of interest is clear from the context.

## 4. Soft Interventions with Latent Variables

In this section, we recall the key ingredients from Kocaoglu et al. (2019) for causal discovery under latents. In this work, each domain corresponds to one soft-interventional regime, i.e., one interventional distribution induced by a known intervention target set.

### 4.1. Do-constraints

**Corollary 4.1** (Mixed do-do / do-see invariance (Kocaoglu et al., 2019, Cor. 1))**.** *Let $\mathcal{D} = (\mathbf{V} \cup \mathbf{L}, E)$ be a causal graph. Fix intervention targets $I, J \subseteq \mathbf{V}$ and disjoint $Y, W \subseteq \mathbf{V}$. Define the symmetric difference $K := I \triangle J$, choose $W_K \subseteq W \cap K$, and let $R := K \setminus W_K$. Let $R(W) \subseteq R$ be any subset of nodes in $R$ that are non-ancestors of $W$ in $\mathcal{D}$. If*

$$Y \perp_{\mathcal{D}_{\underline{W_K}, \overline{R(W)}}} K \mid (W \setminus W_K),$$

*then the following* do-constraint *holds:*

$$P_I(\mathbf{y} \mid \mathbf{w}) \; = \; P_J(\mathbf{y} \mid \mathbf{w}).$$

Do-constraints in Corollary 4.1 mark testable (cross-domain) invariances derived from soft do-calculus and provide a sufficient graphical condition under which two domains agree on the distributional invariance. We define the reverse of it as $c$-faithfulness.

**Definition 4.2** ($c$-faithfulness)**.** *For a causal graph $\mathcal{D} = (\mathbf{V} \cup \mathbf{L}, E)$, a tuple of distributions $(P_I)_{I \in \mathcal{I}}$ is called $c$-faithful to the causal graph if:*

1. *For $I \in \mathcal{I}$, if $P_I(y \mid w, z) = P_I(y \mid w), Y \perp_{\mathcal{D}} Z \mid W$;*

---

[1]Informally, if a variable is intervened upon in multiple regimes, its post-intervention conditional is identical across those regimes.

2. For $I, J \in \mathcal{I}$, if $P_I(y \mid w) = P_J(y \mid w)$, $Y \perp\!\!\!\perp_{\mathcal{D}_{\underline{W_K}, \overline{R(W)}}} K \mid W \setminus W_K$:

Definition 4.2 serves as the basic assumption and can be used as a workhorse of multi-domain causal graph learning.

### 4.2. Augmentation by $F$-nodes

The augmented graph is proposed to capture the graphical conditions of do-constraints without graph mutilations.

**Definition 4.3** (Augmented graph (Kocaoglu et al., 2019, Def. 3)). Let $\mathcal{D} = (\mathbf{V} \cup \mathbf{L}, E)$ and let $\mathcal{I} \subseteq 2^{\mathbf{V}}$ be a set of intervention targets. For each unordered pair of targets $I, J$, construct $\mathbf{F} = \{F_{\{I,J\}}\}$ and the set of edges incident to them $E_F = \{F_{\{I,J\}} \to S, \forall S \in I \Delta J, I, J \in \mathcal{I}\}$. The augmented graph $\mathcal{I}\text{-AUG}(\mathcal{D}) = (\mathbf{V} \cup \mathbf{L} \cup \mathbf{F}, E \cup E_F)$.

To describe Definition 4.3 in words, we assign an $F$ node for each pair of intervention targets and point the $F$ node to the symmetric difference of the targets. We use $F_{\{I,J\}}$ to represent that $F$ node for targets $I, J$ and $tar(F_{\{I,J\}}) = I \Delta J$ to represent the targets it points to.

**Definition 4.4** (Augmented MAG (Kocaoglu et al., 2019, Def. 4)). Given $\mathcal{D}$ and $\mathcal{I}$, the *augmented MAG* or $\mathcal{I}$-MAG is the MAG over the observed node set $\mathbf{V} \cup \mathbf{F}$ obtained by marginalizing latent nodes from the augmented graph:

$$\mathcal{I}\text{-MAG}(\mathcal{D}) := \text{MAG}(\mathcal{I}\text{-AUG}(\mathcal{D})).$$

By taking the MAG[2] of the augmented graph, we preserve all m-separations corresponding to testable distributional invariances. Based on Definition 4.4, we can establish the $\mathcal{I}$-MEC as follows:

**Theorem 4.5** (Characterization of $\mathcal{I}$-MEC (Kocaoglu et al., 2019, Thm. 2)). *Let $\mathcal{D}_1 = (\mathbf{V} \cup \mathbf{L}_1, E_1)$ and $\mathcal{D}_2 = (\mathbf{V} \cup \mathbf{L}_2, E_2)$ be causal graphs, and fix a controlled intervention set $\mathcal{I}$. Define augmented MAGs $\mathcal{M}_1 := \mathcal{I}\text{-MAG}(\mathcal{D}_1)$ and $\mathcal{M}_2 := \mathcal{I}\text{-MAG}(\mathcal{D}_2)$. Then $\mathcal{D}_1$ and $\mathcal{D}_2$ are $\mathcal{I}$-Markov equivalent iff the augmented MAGs $\mathcal{M}_1$ and $\mathcal{M}_2$ satisfy:*

1. *they have the same skeleton;*

2. *they have the same unshielded colliders;*

3. *for any discriminating path shared by both graphs, the discriminated node is a collider on that path in $\mathcal{M}_1$ iff it is a collider on that path in $\mathcal{M}_2$.*

Using the characterization of $\mathcal{I}$-MEC in Theorem 4.5, we can define an objective for any learning algorithms.

---

[2]We use the conventional steps to construct the MAG from an ADMG: For each pair of nodes $X, Y$, if $X$ is $Y$'s ancestor/descendant/spouse and there is an inducing path between them, we orient $X \to Y / X \leftarrow Y / X \leftrightarrow Y$ between them, otherwise they are not adjacent.

**Definition 4.6** ($\mathcal{I}$-PAG). Given a causal graph $\mathcal{D}$ and a set of intervention targets $\mathcal{I}$, let $\mathcal{M} = \mathcal{I}\text{-MAG}(\mathcal{D})$ and let $[\mathcal{M}]$ be the set of $\mathcal{I}$-MAGs corresponding to all the causal graphs that are $\mathcal{I}$-Markov equivalent to $\mathcal{D}$ given $\mathcal{I}$. The $\mathcal{I}$-PAG, denoted as $\mathcal{P}(\mathcal{D}, \mathcal{I})$, is a graph such that:

1. $\mathcal{P}(\mathcal{D}, \mathcal{I})$ has the same adjacencies as $\mathcal{M}$, and any member of $[\mathcal{M}]$ does; and

2. every non-circle mark in $\mathcal{P}(\mathcal{D}, \mathcal{I})$ is shared across all members in $[\mathcal{M}]$; and

3. every circle mark in $\mathcal{P}(\mathcal{D}, \mathcal{I})$ is not invariant: for each circle endpoint, there exist two members of $[\mathcal{M}]$ in which that endpoint is an arrowhead and an arrowtail, respectively.

Accordingly, Kocaoglu et al. (2019) propose an FCI-inspired algorithm $\mathcal{I}$-FCI to learn the $\mathcal{I}$-PAG associated with $\mathcal{I}\text{-MAG}(\mathcal{D})$ by combining CI tests and do-constraint tests. The algorithm is sound under $c$-faithfulness, but its orientation closure is not complete, and thus cannot output the $\mathcal{I}$-PAG in general. Our contribution will be to add additional orientation principles to decrease this gap.

## 5. New Rules and the Learning Algorithm

In this section, we present our insights into the $\mathcal{I}$-MAG structure, a learning algorithm through MAG listing, and the new orientation rules for efficient learning. Without loss of generality, we treat an underlying causal DAG with latent variables as equivalently represented by its latent projection ADMG on $\mathbf{V}$, and work directly with this ADMG.

### 5.1. Realization of $\mathcal{I}$-MAG

We have discussed a key observation that an $F$ node $F_{\{I,J\}}$ can be adjacent to nodes that are not members of $tar(F_{\{I,J\}})$ in the $\mathcal{I}$-MAG through an inducing path via some targets. We call any $Y \notin tar(F)$ but is adjacent to $F$ in the $\mathcal{I}$-MAG a non-target node of $F$ and the $(F, Y)$ edge a non-target edge. Based on this, we introduce a useful lemma that characterizes such inducing paths.

**Lemma 5.1.** *If a non-target node $Y$ of $F_{\{I,J\}}$ is adjacent to $F_{\{I,J\}}$ in an $\mathcal{I}$-MAG of $\mathcal{D}$, then there is an inducing path from $F_{\{I,J\}}$ to $Y$ in $\mathcal{I}\text{-AUG}(\mathcal{D})$. Additionally, there is a target $X \in tar(F_{\{I,J\}})$ such that there is an inducing path from $X$ to $Y$ that starts with an arrowhead at $X$, and a directed path from $X$ to $Y$ in $\mathcal{I}\text{-AUG}(\mathcal{D})$.*

Lemma 5.1 establishes the graphical condition for each non-target node to be adjacent to an $F$ node in $\mathcal{I}$-MAG. It can be used to infer the ancestral relationship and existence of backdoor paths from the $F$ node adjacency. Inspired by this, we introduce the following definitions.

**Algorithm 1** $\mathcal{I}$-MAG Realizability

**Input:** mixed graph $\mathcal{M}$ on $\mathbf{V} \cup \mathbf{F}$, intervention targets $\mathcal{I}$
**Output:** `true` and a witness assignment $\tau$, a realizing ADMG $\mathcal{D}$, or `false`
**if** BASICFCHECKS$(\mathcal{M}, \mathcal{I}) = $ `false` **then**
    **return** `false`
**end if**
$\mathcal{D}_0 \leftarrow \mathcal{M}[\mathbf{V}]$
$\mathbf{P} \leftarrow \{(F, Y) : F \in \mathbf{F},\ Y \in Adj_{\mathcal{M}}(F) \setminus (tar(F) \cup \{F\})\}$
$(Cand, \mathbf{P}') \leftarrow $ BUILDCANDIDATES$(\mathcal{M}, \mathcal{D}_0, \mathcal{I}, \mathbf{P})$
**if** $Cand = $ `fail` **then**
    **return** `false`
**end if**
$(ok, \tau, \mathcal{D}) \leftarrow $ SEARCHTRANSIT$(\mathcal{M}, \mathcal{D}_0, \mathcal{I}, Cand, \mathbf{P}')$
**if** $ok$ **then**
    **return** `true`, $\tau, \mathcal{D}$
**else**
    **return** `false`
**end if**

---

**Definition 5.2** (Transit sets/nodes). Let $\mathcal{D}$ be an underlying ADMG on $\mathbf{V}$ and let $\mathcal{I}$-AUG$(\mathcal{D})$ be the augmented graph with edges $F_{\{I,J\}} \to X$ for all $X \in K = I \Delta J$. Consider $Y$ as a non-target node of $F_{\{I,J\}}$ in $\mathcal{I}$-MAG$(\mathcal{D})$. A node $X \in K$ is called a *transit node* for the pair $(F_{\{I,J\}}, Y)$ in $\mathcal{I}$-AUG$(\mathcal{D})$, if there exists an *inducing path* $\pi$ between $F_{\{I,J\}}$ and $Y$ in $\mathcal{I}$-AUG$(\mathcal{D})$ such that the neighbor of $F_{\{I,J\}}$ on $\pi$ is $X$. We say $(X, Y)$ is a *transit pair*. The set of all transit nodes for the pair $(F_{\{I,J\}}, Y)$ is called a transit set denoted as $Tr(F_{\{I,J\}}, Y)$.

Transit nodes can be read from the augmented graphs by checking the inducing paths between $F$ nodes and their non-target nodes. The following lemma tells us how to construct an ADMG that has the same $\mathcal{I}$-MAG as a given ADMG and a target set using transit sets.

**Lemma 5.3.** *For an arbitrary ADMG $\mathcal{D} = (\mathbf{V}, E)$, construct its $\mathcal{I}$-MAG $\mathcal{M} = \mathcal{I}$-MAG$(\mathcal{D})$ with respect to a set of targets $\mathcal{I} \subseteq 2^V$. We can construct another ADMG $\mathcal{D}' = (\mathbf{V}, E')$ such that $\mathcal{M} = \mathcal{I}$-MAG$(\mathcal{D}')$ with the following process: (1) Let $\mathcal{D}' = \mathcal{M}[\mathbf{V}]$ be the subgraph on $\mathbf{V}$, construct $\mathcal{I}$-AUG$(\mathcal{D}) = \mathcal{A}$ and $\mathcal{I}$-AUG$(\mathcal{D}') = \mathcal{A}'$. (2) In $\mathcal{A}$, for each pair of targets $I, J \in \mathcal{I}$, find all transit pairs $\{(X, Y)\}$ for $F_{\{I,J\}}$. For each such valid transit pair, add a bidirected edge between $X, Y$ in $\mathcal{D}'$ if the bidirected edge $X \leftrightarrow Y$ is not already present.*

Lemma 5.3 shows that to recover some representative ADMG from the $\mathcal{I}$-MAG, we just need its observational MAG and the transit pairs. This is straightforward if the $\mathcal{I}$-AUG is given. However, when we do not have access to $\mathcal{I}$-AUG but only the $\mathcal{I}$-MAGs, we can just identify a set of nodes to be the possible transit nodes based on the graphical properties of transit nodes. Obviously, such nodes have to

be parents of the non-target node of interest.

**Definition 5.4** (Candidate transit nodes). Let $\mathcal{M}$ be an $\mathcal{I}$-MAG and let $Y$ be a non-target node of $F$. The set of *candidate transit nodes* for the pair $(F, Y)$ is

$$Cand_{\mathcal{M}}(F, Y) := tar(F) \cap Pa_{\mathcal{M}}(Y).$$

The true transit node is guaranteed to be in the candidate set accordingly to Lemma 5.1. Therefore, given the $\mathcal{I}$-MAG, we can use a search algorithm to find out possible transit pairs to cover all non-target edges. Nevertheless, it should return a null result when no appropriate transit pairs that explain all the non-target edges can be found, when the given mixed graph is not a valid $\mathcal{I}$-MAG. Accordingly, we propose the following lemma to characterize an $\mathcal{I}$-MAG.

**Lemma 5.5** ($\mathcal{I}$-MAG characterization). *Given an arbitrary mixed graph $\mathcal{M} = (\mathbf{V} \cup \mathbf{F}, E)$ and a target set $\mathcal{I} \subseteq 2^{\mathbf{V}}$, $\mathcal{M}$ is a valid $\mathcal{I}$-MAG, if and only if there exists a set of transit pairs, such that by adding bidirected edges at these pairs to $\mathcal{M}[\mathbf{V}]$, named as $\mathcal{M}'$, $\mathcal{I}$-MAG$(\mathcal{M}') = \mathcal{M}$.*

Based on Lemma 5.3, Lemma 5.5 claims that we can verify if a mixed graph is an $\mathcal{I}$-MAG by checking if we can construct an ADMG with the same mixed graph as its $\mathcal{I}$-MAG. We can systematically check the property by finding if such transit pairs exist. As we have shown, there are mixed graphs that follow all FCI constraints but are not $\mathcal{I}$-MAGs. Such constraint beyond FCI rules relate closely to the $F$ node semantics and can be affected by the global structure. To illustrate this, we show 2 examples in Figure 2. One example shows that transit nodes cannot be selected independently. The second example shows that a global structure may restrict some targets to be a transit node. Therefore, if only a mixed graph is given, we need a process to find the transit nodes to cover all non-target edges.

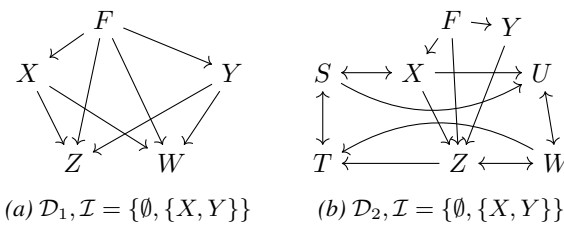

(a) $\mathcal{D}_1, \mathcal{I} = \{\emptyset, \{X, Y\}\}$      (b) $\mathcal{D}_2, \mathcal{I} = \{\emptyset, \{X, Y\}\}$

*Figure 2.* An example to show that the identification of valid transit pairs requires checking of the global structure. In Figure 2a, we need to assign the transit nodes for $Z, W$. Since $X, Y$ are both parents of them, either of them can be the transit node for $Z, W$. However, either of them cannot be the transit node for both $Z, W$, because otherwise, it will create an inducing path between $Z, W$ but they are not adjacent in $\mathcal{D}_1$. In Figure 2b, we need to find the transit node for $Z$. $X, Y$ are both parents of $Z$. However, if we add a bidirected edge between $X, Z$, it will create an inducing path between $U, T$ while they are not adjacent in $\mathcal{D}_2$. Hence, $\mathcal{D}_2$ is not a valid $\mathcal{I}$-MAG of any ADMG.

**Definition 5.6** (Transit selection set). Let $\mathcal{M}$ be a mixed graph with $F$-nodes. Define the global target set $K :=$

**Algorithm 2** MAG-listing-based $\mathcal{I}$-PAG completion

---

**Input:** targets $\mathcal{I}$, interventional distributions $(P_I)_{I \in \mathcal{I}}$, observables $\mathbf{V}$
**Output:** maximally oriented augmented PAG $\widehat{\mathcal{P}}$
$\mathcal{P}_0, \mathbf{F} \leftarrow \mathcal{I}\text{-FCI-BASE}(\mathcal{I}, (P_I)_{I \in \mathcal{I}}, \mathbf{V})$
For each endpoint $e$ in $\mathcal{P}_0$, let $Mark(e) = \emptyset$
**for** $\mathcal{M}_F \in \text{MAGLIST}(\mathcal{P}_0)$ **do**
    **if** IMAG-REALIZE$(\mathcal{M}_F, \mathcal{I}) = \texttt{true}$ **then**
        $Mark \leftarrow \text{INTERSECTMARKS}(\mathcal{M}_F, Mark)$
    **end if**
**end for**
Insert marks in $Mark$ into $\mathcal{P}_0$ to get $\widehat{\mathcal{P}}$
**return** $\widehat{\mathcal{P}}$

---

$\bigcup_F tar(F)$, and $U := \{(F,Y) : Y \in Adj_{\mathcal{M}}(F) \setminus (tar(F) \cup \{F\})\}$, i.e., $U$ is the set of non-target edges in $\mathcal{M}$. For each $(F,Y) \in U$, let $Cand_{\mathcal{M}}(F,Y) \subseteq tar(F)$ be the candidate witness set computed from $\mathcal{M}$. A *transit selection set* is any set $S \subseteq K \times U$. We say that $S$ *covers* $U$ if for every $(F,Y) \in U$ there exists $X \in Cand_{\mathcal{M}}(F,Y)$ such that $(X, (F,Y)) \in S$.

Equivalently, $S$ induces a *witness assignment* $\tau : U \to 2^K$ defined by $\tau(F,Y) := \{X \in tar(F) : (X, (F,Y)) \in S\}$. We call $\tau$ *valid* if $\tau(F,Y) \subseteq Cand_{\mathcal{M}}(F,Y)$ for all $(F,Y) \in U$. A *single-witness assignment* is a map $\tau : U \to K$ such that $\tau(F,Y) \in Cand_{\mathcal{M}}(F,Y)$, for all $(F,Y) \in U$.

To search for a valid transit selection set, we propose Algorithm 1. For an arbitrary input mixed graph $\mathcal{M}$, it enumerates all possible candidates of transit nodes for each $(F,Y)$ pair, therefore a valid set can be found as long as the input mixed graph is a valid $\mathcal{I}$-MAG. To make the searching more efficient, we start from a non-target $Y$ with the minimum number of candidate transit nodes and perform a DFS for each $(F,Y)$ pair. For each selection set it picks, we will add the corresponding bidirected edges to $\mathcal{M}[\mathbf{V}]$ and check if its $\mathcal{I}$-MAG is the same as $\mathcal{M}$.

**Theorem 5.7.** *A mixed graph $\mathcal{M}$ is a valid $\mathcal{I}$-MAG given an intervention target set $\mathcal{I} \subseteq 2^V$, if and only if the output of Algorithm 1 is* `true`.

Algorithm 1 provides a systematic procedure for verifying whether a candidate mixed graph is a valid $\mathcal{I}$-MAG, and can therefore be used as a subroutine in our learning algorithm. While Wang et al. (2025) give an efficient method for listing MAGs consistent with a given PAG, Figure 2 shows that not every MAG containing $F$ nodes is a valid $\mathcal{I}$-MAG. Accordingly, we enumerate candidate MAGs and retain only those that pass the validity test in Algorithm 1.

Algorithm 2 has two major phases. In the first phase, we learn the skeleton of the $\mathcal{I}$-PAG, apply FCI rules, and orient edges out of $F$ nodes to get $P_0$ as an FCI closure. This is similar to $\mathcal{I}$-FCI but without applying $\mathcal{I}$-FCI 's extra rule. The outcome $\mathcal{P}_0$ at this stage can be considered as a PAG

with local BK at $F$ nodes that edges incident to $F$ nodes go outwards and maximally oriented under FCI rules. Without $F$ node semantics, $\mathcal{P}_0$ is sound and complete as shown in Kocaoglu et al. (2019) and Wang et al. (2022). Therefore, the next phase of the algorithm is to orient any marks that are shared across all valid $\mathcal{I}$-MAGs using a MAG listing algorithm MAGLIST. For this, we can use the efficient MAG listing algorithm in Wang et al. (2025). It can list any valid MAG given a PAG. With any listed MAG, we call Algorithm 1 to check if it is valid $\mathcal{I}$-MAG and record its edge marks if it is valid. Finally, we insert the edge marks that are shared across all valid $\mathcal{I}$-MAGs identified to $\mathcal{P}_0$ to get the output $\widehat{\mathcal{P}}$. Assuming we check all valid MAGs for $\mathcal{P}_0$ as the partially oriented graph, we will not miss any valid $\mathcal{I}$-MAGs and thus Algorithm 2 is in theory sound and complete. However, it can be computationally prohibitive in the worst case due to the number of consistent MAGs.

**Theorem 5.8.** *Algorithm 2 is sound and complete assuming exhaustive $\mathcal{I}$-MAG completion enumeration.*

### 5.2. New Orientation Rules

Although Algorithm 2 outputs a maximally oriented graph, it relies on the MAG listing algorithm which can be computationally expensive when the number of MAG completions is large. Here we aim to propose an efficient approach that requires only local checks. The key is if we can identify the transit nodes without searching globally. Once we identify the transit nodes, we can incorporate the structural constraints related into the graph as a set of orientation rules. Inspired by this observation, we propose the following 3 new orientation rules as an extension of $\mathcal{I}$-FCI . The new rules are not complete, but provide an efficient local approach.

**Rule 9.** For a node $Y$ that is adjacent to $F$ and $Y \notin tar(F)$, if it has only one possible parent that is a target node $X$, $X \in tar(F)$, orient $X \ast\!\!-\!\!\ast Y$ as $X \to Y$, record $(X, Y)$ as a transit pair.

**Rule 10.** For a node $Z$ adjacent to $F$ and also adjacent to a target $X \in tar(F)$, if there is some non-target $Y \notin tar(F)$ adjacent to $F$, but $Y$ is not adjacent to $Z$, and $(X, Y)$ is a transit pair, orient $Z \ast\!\!-\!\!\ast X$ as $Z \leftarrow X$.

**Rule 11.** For a non-target node $Y \notin tar(F)$ adjacent to $F$, if it is adjacent to multiple targets of $F$ with $X$ as the only source node of the directed target-induced subgraph, then orient $X \ast\!\!-\!\!\ast Y$ as $X \to Y$.

Rule 9 is an extension of Rule 9 of $\mathcal{I}$-FCI . The original rule says that if the $F$ node adjacency has only one target, we can orient edges going from the target to non-target nodes. Our new rule extends to multi-target cases. According to Lemma 5.1, if a non-target node is adjacent to $F$, there

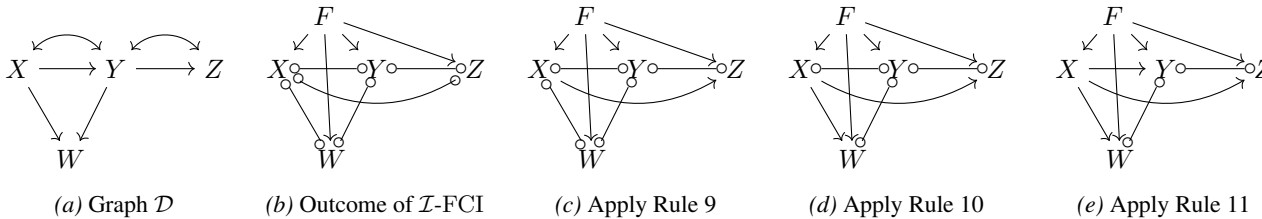

*(a)* Graph $\mathcal{D}$     *(b)* Outcome of $\mathcal{I}$-FCI     *(c)* Apply Rule 9     *(d)* Apply Rule 10     *(e)* Apply Rule 11

*Figure 3.* An example to show that our proposed new orientation rules can recover more edges. Figure 3a is the ground truth graph with intervention targets $\mathcal{I} = \{\emptyset, \{X, W\}\}$. Figure 3b is the learning outcome by $\mathcal{I}$-FCI . No rules in $\mathcal{I}$-FCI can be applied. Figure 3c shows the graph after applying Rule 9 which orients $X \to Z$. Figure 3d shows the graph after applying Rule 10 which orients $X \to W$. Figure 3e shows the graph after applying Rule 11 which orients $X \to Y$.

exists an inducing path and a directed path from a target to the non-target, thus the non-target node has to be a child of the target in the $\mathcal{I}$-MAG. We orient the edge whenever other targets adjacent cannot be a parent of the non-target node in the partially oriented graph and recognize this node as a transit node for the non-target node. Nevertheless, the identified transit pair indicates that there is an inducing path which can potentially break the maximality of the graph. Therefore, we introduce Rule 10 to avoid this. Whenever we recover a transit pair $(X, Y)$, an edge $Z* \to X$ would create an inducing path from $Z$ to $Y$ via $X$ if we put an arrowhead at $X$ from $Z$. This is prohibited when $Z, Y$ are not adjacent, and thus we can orient $Z \leftarrow X$. Rule 11 states that when a non-target is adjacent to multiple targets, it has to be a child of the target with the highest topological order. To witness, we know that there has to be an inducing path from (at least) one target to the non-target. If the non-target is accessed by $F$ through any lower-order target, the ancestrality will force it to be the child of the highest-order target.

To illustrate how the new rules work, we present an example in Figure 3. Concretely, Figure 3a is the ground truth graph with intervention targets $\mathcal{I} = \{\emptyset, \{X, W\}\}$. Figure 3b is the learning outcome by $\mathcal{I}$-FCI . The only edges $\mathcal{I}$-FCI can recover are those out of $F$. However, we notice that $Z$ as a non-target is only adjacent to one target $X$, therefore Rule 9 will orient $X \to Z$. Rule 9 also indicates an inducing path from $X$ to $Z$, and thus Rule 10 will be triggered to orient $X \to W$ since $W$ and $Z$ are not adjacent and there cannot be any inducing path between them. Finally, Rule 11 is triggered to orient $X \to Y$. To witness, we can infer that there is an inducing path from $F$ to $Y$ through $X$ or $Z$. If it is through $X$, it implies that $X \to Y$. Otherwise, it is through $Z$ indicating an inducing path from $Z$ to $Y$. This immediately creates another inducing path from $X$ to $Y$ through $Z$. Consequently, either way, we can infer that $X \to Y$. No rules can be applied any further.

Notice that all our proposed new rules apply only to nodes adjacent to $F$ nodes. The insight is that the $F$ node adjacency encodes the structural constraints on the underlying ADMG, which is more informative than just applying local

BK as defined in Wang et al. (2022) around $F$ nodes. This is also why the new rules are not covered by FCI rules.

**Theorem 5.9.** *The 3 new orientation rules are sound.*

Although efficient, the new rules are not complete. The current Rule 9 can only identify transit nodes by checking local properties. However, there can be cases that a transit node has to be identified by checking global conditions as illustrated in Figure 2. Given these challenges, we stick to the local efficient rules for this work.

## 6. Experiments

We compare $\mathcal{I}$-FCI (Kocaoglu et al., 2019), LIST, and FAST on randomly generated ADMGs. For each setting, we generate random ADMGs with $n = 5$ observed variables. We first sample a random topological order and generate directed edges with density $\rho_{\mathrm{DAG}} = 0.5$. We then independently add bidirected edges with latent-confounding density $\rho_{\mathrm{bi}} \in \{0.5, 0.8\}$. Given the generated ADMG, we construct binary Bayesian networks using random CPTs and generate interventional datasets with 50,000 samples per domain using pyAgrum (Ducamp et al., 2020). We use three domains, one of which is observational, and intervention targets have maximum size 2. If a variable appears in multiple intervention targets, we impose the same shifted mechanism for that variable across those domains, matching the controlled soft-intervention assumption.

For evaluation, we compare each learned augmented graph against the ground-truth augmented MAG. Specifically, for each sampled ADMG, we first construct its augmented graph by attaching the corresponding $F$-nodes to the symmetric differences of intervention targets, and then take the MAG of this augmented graph. We report partial-graph SHD (PG-SHD), which counts adjacency and endpoint-mark disagreements between the learned augmented graph and the ground-truth augmented MAG. We also report endpoint F1 (End-F1), which measures recovery of endpoint marks in the augmented MAG. Lower PG-SHD and higher End-F1 are better. We report results on 100 repeated trials

| **Alg.** | $n = 5, \rho_{\mathrm{bi}} = 0.5$ | | $n = 5, \rho_{\mathrm{bi}} = 0.8$ | |
|---|---|---|---|---|
| | PG-SHD↓ | END-F1↑ | PG-SHD↓ | END-F1↑ |
| $\mathcal{I}$-FCI | $11.21\pm4.70$ | $0.771\pm0.097$ | $14.23\pm4.90$ | $0.732\pm0.102$ |
| FAST | $\mathbf{10.87\pm4.69}$ | $\mathbf{0.777\pm0.098}$ | $\mathbf{13.90\pm5.08}$ | $\mathbf{0.738\pm0.106}$ |
| LIST | $11.04\pm4.82$ | $0.774\pm0.100$ | $14.16\pm5.02$ | $0.733\pm0.105$ |

*Table 1.* Comparison across random ADMGs with $n = 5$ observed variables and different latent-confounding densities. PG-SHD compares the learned augmented graph with the ground-truth augmented MAG, while End-F1 measures endpoint-mark recovery. Lower PG-SHD and higher End-F1 are better.

and report the mean and standard error.

Table 1 shows that FAST consistently improves over $\mathcal{I}$-FCI in both PG-SHD and End-F1, while preserving the same skeleton learned by the baseline. This supports the role of the additional local rules: they do not change the skeleton-learning phase, but can orient additional endpoint marks implied by the $F$-node semantics. LIST also improves over $\mathcal{I}$-FCI in the $n = 5, \rho_{\mathrm{bi}} = 0.5$ setting, but its gains are smaller and it is less robust under finite-sample errors, since it enforces global $\mathcal{I}$-MAG realizability constraints. Overall, FAST provides the best empirical tradeoff among the three methods in these simulations.

## 7. Conclusion

We studied causal discovery from *controlled soft interventions* in the presence of latent variables under the augmented-graph framework of Kocaoglu et al. (2019). While the augmented MAG characterization precisely describes the interventional Markov equivalence class, existing FCI-inspired learning procedures are not complete. To decrease this gap, we identified additional structural information carried by $F$-node adjacency and their implications for inducing paths through interventional targets. Building on this insight, we establish the realizability of $\mathcal{I}$-MAGs to systematically judge if a given mixed graph is a valid $\mathcal{I}$-MAG.

Inspired by the $\mathcal{I}$-MAG realizability we design two approaches to extend the $\mathcal{I}$-FCI learning process. We introduced three new local orientation rules and integrated them into $\mathcal{I}$-FCI . The rules are sound and efficient. We further design an enumeration-based completion procedure using MAG listing. Although it is sound and theoretically complete, its worst-case runtime can be prohibitive since there can be super-exponentially many MAG completions consistent with the learned partial graph.

Empirically, our method reduces the number of undetermined edge marks relative to $\mathcal{I}$-FCI on the simulated data. Our formal results focus on controlled soft interventions with known observed targets. Extending the approach to hard interventions is nontrivial because hard interventions remove incoming edges into targets and therefore induce different separation constraints. Unknown-target soft interven-

tions are handled by the complementary $\psi$-FCI framework, while our setting exploits the additional target-mediated structure available when intervention targets are known. Interventions on latent variables are also outside our formal guarantees, although in applications they may sometimes be approximated by downstream observed mechanism changes.

An interesting direction for future work is to design an efficient complete learning algorithm without MAG listing.

## Impact Statement

This paper advances methodological foundations for causal discovery with latent confounders when data are collected under multiple soft-interventional regimes. By improving identifiability (i.e., provably orienting additional edge endpoints that are already determined by the available invariances), our results can strengthen causal analyses in scientific and engineering workflows that use perturbation experiments or multi-environment data, potentially leading to more reliable hypothesis generation and experimental design. At the same time, causal discovery methods can be misapplied in high-stakes settings (e.g., healthcare, policy, or automated decision systems) if their assumptions—such as faithfulness, correct intervention specification, and sufficient sample sizes for invariance testing—are violated; such misuse could contribute to incorrect causal conclusions and downstream harm. The same perspective may also inform biological diagnostic workflows, such as marker-gene or gene-panel prioritization in interpretable single-cell analysis (Zhou et al., 2022; Plumb et al., 2020; Lu et al., 2021), though such applications would require domain-specific validation.Our contribution is theoretical and algorithmic and does not introduce new data collection or deployment mechanisms. We therefore emphasize that the proposed guarantees are conditional on the stated assumptions, and that practitioners should combine such tools with domain knowledge, careful diagnostic checks, and sensitivity analyses when drawing consequential conclusions.

## Acknowledgement

This research has been supported in part by NSF CAREER 2239375, IIS 2348717, Amazon Research Award, Adobe

Research and Intuit..

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

# Appendix Contents

## A. Detailed Related Works

**Learning from combined datasets.** A large literature studies causal discovery from multiple experimental conditions or domains, but many approaches are primarily empirical and rely on additional modeling assumptions. Representative lines of work assume Markovianity or specific functional forms (e.g., linearity), and consider mixtures of observational and interventional data with known or unknown targets (Peters et al., 2016; Ghassami et al., 2017; Heinze-Deml et al., 2018; Huang et al., 2020; Perry et al., 2022). The Joint Causal Inference (JCI) framework pools data across contexts by augmenting the variable set with context indicators and then performs discovery on the combined dataset (Mooij et al., 2020). Neural and differentiable frameworks have also been proposed to leverage combined observational and interventional data and can perform well empirically even with unknown targets, but typically without accompanying soundness guarantees for the returned graph object (Ke et al., 2019; Brouillard et al., 2020; Lopez et al., 2022).

Recent work has started to analyze more realistic intervention imperfections, including *off-target* interventions (Choo et al., 2024) and *mixtures* of unknown interventions (Kumar et al., 2024). Beyond single-system DAG discovery, interventions have also been studied for identifying structure in *mixtures of DAGs* (Varıcı et al., 2024b), and for causal representation learning under unknown *multi-node* intervention environments (Varıcı et al., 2024a).

On the theoretical side, Acharya et al. (2018) give sample-complexity bounds for testing whether an unknown causal Bayesian network matches a reference model under interventions, and Jiang & Aragam (2023) characterize conditions for identifiability of causal representations without parametric assumptions, even with unknown interventions and without assuming faithfulness. Orthogonal to estimation, intervention design work seeks to choose a small or low-cost set of interventions to orient causal/ancestral relations (Addanki et al., 2021; Tigas et al., 2022), including recent adaptive designs (Elahi et al., 2024). Finally, under causal sufficiency and known targets, score-based methods such as GIES and IGSP aim to return a single DAG rather than an explicit equivalence class representation (Hauser & Bühlmann, 2012; Wang et al., 2017), and Bayesian approaches have also been explored in this regime (Mascaro & Castelletti, 2023; Zhou et al., 2024).

## B. Proofs

### B.1. Proof for Lemma 5.1

*Proof.* Since $F$ and $Y$ are adjacent in the MAG $\mathcal{M} = \text{MAG}(\mathcal{I}\text{-AUG}(\mathcal{D}))$, by maximality there exists an inducing path $\pi = \langle F, V_1, \ldots, V_k, Y \rangle$ between $F$ and $Y$ in $\mathcal{A} = \mathcal{I}\text{-AUG}(\mathcal{D})$. Because $F$ is a source node in $A$ which has no incoming edges by construction, $F$ cannot have an arrowhead incident to it. Hence the first edge on $\pi$ must be $F \to V_1$. Let $X := V_1$. By construction, $X \in tar(F)$.

All non-endpoint vertices on an inducing path are colliders. Since $X$ is not an endpoint of $\pi$, it is a collider on $\pi$. Because the edge $F \to X$ contributes an arrowhead into $X$, collider status of $X$ implies that the second edge on $\pi$ also has an arrowhead at $X$.

By definition of inducing path, every collider on $\pi$ is an ancestor of at least one endpoint of $\pi$. No vertex can be an ancestor of $F$ since $F$ is a source node. Therefore every collider on $\pi$ is an ancestor of $Y$. In particular, $X \in An_{\mathcal{A}}(Y)$, and thus there exists a directed path $X \to \cdots \to Y$ in $\mathcal{A}$. Since all edges out of $F$ point into observed target variables, this directed path lies entirely in $\mathbf{V}$ and hence is also a directed path in the underlying ADMG $\mathcal{D}$.

Finally, consider the suffix path $\pi[X \rightsquigarrow Y] = \langle X, V_2, \ldots, V_k, Y \rangle$. Every internal vertex of this suffix is also an internal vertex of $\pi$, hence a collider and an ancestor of $Y$. Thus $\pi[X \rightsquigarrow Y]$ is an inducing path between $X$ and $Y$ in $A$. Moreover, the first edge on this suffix has an arrowhead into $X$, completing the proof. □

### B.2. Proof for Lemma 5.3

*Proof.* After first step of initializing $\mathcal{D}'$ with $\mathcal{M}[V]$, the only difference between $\mathcal{I}\text{-MAG}(\mathcal{D})$ and $\mathcal{I}\text{-MAG}(\mathcal{D}')$ is $F$ nodes' adjacency. We show this with the following lemma.

**Lemma B.1.** *For an ADMG $\mathcal{D}$ with $\mathcal{I}\text{-MAG}(\mathcal{D}) = \mathcal{M}$ w.r.t. $\mathcal{I}$, then for $\mathcal{D}' = \mathcal{M}[\mathbf{V}]$, $\mathcal{I}\text{-MAG}(\mathcal{D}') = \mathcal{M}'$, $\mathcal{M}$ and $\mathcal{M}'$ have the same induced graph on $\mathbf{V}$.*

*Proof.* Construct the augmented graph $\mathcal{A}' = \mathcal{I}\text{-AUG}(\mathcal{D}')$. We need to show that adding $F$ nodes to $\mathcal{M}'$ will not create any inducing path between non-adjacent pair of nodes. For any pair of nodes in $\mathbf{V}$, since $\mathcal{M}'$ is maximal, any path that

does not go through $F$ nodes between two non-adjacent nodes cannot be an inducing path. On the other hand, any path that goes through $F$ nodes cannot be an inducing path either since $F$ nodes have only outward edges and thus cannot be a collider. □

In $\mathcal{M}$, some $F_{(I,J)}$ is adjacent to $Y \notin K, K = I\Delta J$ indicates that there is an inducing path from $F_{(I,J)}$ to $Y$ that goes through some $X \in K$ in $\mathcal{I}$-AUG$(\mathcal{D})$. Since any node on the path cannot be an ancestor of $F_{(I,J)}$, it has to be an ancestor of $Y$. Therefore, $X$ is an ancestor of $Y$ in $\mathcal{D}$. All the nodes between $X, Y$ are colliders in $\mathcal{D}$. Thus the subpath from $X$ to $Y$ is also an inducing path and thus $X \to Y$ appears in $\mathcal{D}'$. By adding $X \leftrightarrow Y$ to $\mathcal{D}'$, when we construct augmented graph of $\mathcal{D}'$, we create an inducing path from $F_{(I,J)}$ to $Y$ and thus we have $F_{(I,J)} \to Y$ in $\mathcal{I}$-MAG$(\mathcal{D}')$. Accordingly, for any non-target node that is adjacent to $F$ nodes in $\mathcal{M}$, it will be adjacent to the same $F$ nodes in $\mathcal{M}'$. What is left to show is that we do not create any extra inducing path between non-adjacent nodes in $\mathcal{I}$-AUG$(\mathcal{D}')$ after step 2.

For the sake of contradiction, consider an inducing path $p$ between non-adjacent $S, T$ in $\mathcal{A}' = \mathcal{I}$-AUG$(\mathcal{D}')$ that goes through a minimal number of transit pairs $(X_1, Y_1), \ldots, (X_k, Y_k)$. We first show that there is also an inducing path between $S, T$ in $\mathcal{A} = \mathcal{I}$-AUG$(\mathcal{D})$.

According to the definition of inducing path, every internal node on $p$ is a collider and an ancestor of either $S$ or $T$. The fact that $(X, Y)$ are adjacent on $p$ indicates that there is an inducing path $p_{XY}$ between $X, Y$ in $\mathcal{A}$. Similarly, every internal node on $p_{XY}$ is a collider and an ancestor of $X$ or $Y$. Since we have $X, Y$ to be an ancestor of $S$ or $T$, the internal nodes on $p_{XY}$ are also ancestors of $S$ or $T$. For each internal node $X$ on $p$, if it is not adjacent to any transit pair, it has to be a collider on $p$. Otherwise, it is adjacent to at least one transit pair. If $X, Y$ are adjacent in a MAG, with an arrowhead at $X$, it is obvious that there is an inducing path with an arrowhead at $X$ between $X, Y$ in the original graph. Therefore if a node on $p$ is not adjacent to any transit pair, it is a collider on the path that concatenates the two inducing paths with it as an end node in $\mathcal{A}$. The rest is to show that if a node is adjacent to transit pairs, it is also a collider that concatenates two adjacent inducing paths in $\mathcal{A}$.

If $X$ is adjacent to two transit pairs $(X_{i-1}, X_i)$ and $(X_i, X_{i+1})$, then $X = X_i$. If $X$ is not a collider, this indicates that there exists $X_{i-1} \to X_i \to X_{i+1}$ (or $X_{i-1} \leftarrow X_i \to X_{i+1}$ or $X_{i-1} \leftarrow X_i \leftarrow X_{i+1}$) and $X_{i-1} \leftrightarrow X_i \leftrightarrow X_{i+1}$ in $\mathcal{A}'$. This immediately creates an inducing path between $X_{i-1}, X_{i+1}$ and thus we can construct another inducing path $p'$ between $S, T$ that skips $X_i$ which goes through less transit pairs. This is a contradiction.

Therefore, there cannot be two consecutive transit pairs on $p$ unless $X$ is a collider. Consider a transit pair $(X_i, X_{i+1})$ on $p$ with $X_i \to X_{i+1}$ and $X_i \leftrightarrow X_{i+1}$. Since $X_i$ is a collider on $p$, the node before $X_i$ is incident to an edge with arrowhead at $X_i$. According to Lemma 5.1, in $\mathcal{A}$, there is an inducing path $q$ between $X_i, X_{i+1}$ that starts with an arrowhead at $X_i$. Hence $X_i$ is also a collider when concatenating the adjacent inducing paths in $\mathcal{A}$. Every internal node on $q$ is a collider and an ancestor of $X_i$ or $X_{i+1}$. Since $X_i$ and $X_{i+1}$ are ancestors of $S$ or $T$, all the internal nodes on $q$ are also ancestors of $S$ or $T$. By replacing the segment $X_i, X_{i+1}$ on $p$ with $q$ as $p'$. We have shown that each segment on $p$ corresponds to an inducing path between the two endpoints in $\mathcal{A}$. By concatenating them in the same order, we get a path $p_{\mathcal{A}}$. Each internal node on $p_{\mathcal{A}}$ is a collider and an ancestor of $S$ or $T$. Also by the definition of $p$, each node on $p$ is also a collider and an ancestor of $S$ or $T$ on $p_{\mathcal{A}}$. Therefore, under this assumption, there has to be an inducing path between $S$ and $T$ in $\mathcal{A}$. This is a contradiction.

If $p$ does not go through any transit pair, according to the maximality of $\mathcal{D}'$ and $\mathcal{A}'$, $p$ cannot be an inducing path. Thus, we show adding bidirected edges between transit pairs will not construct inducing path between non-adjacent nodes in $\mathcal{A}'$.

□

### B.3. Proof for Lemma 5.5

*Proof.* ($\Rightarrow$) Suppose $\mathcal{M}$ is a valid $\mathcal{I}$-MAG for targets $\mathcal{I}$. Then there exists an ADMG $\mathcal{D}$ on $\mathbf{V}$ such that $\mathcal{I}$-MAG$(\mathcal{D}) = \mathcal{M}$. Consider the augmented graph $\mathcal{A} := \mathcal{I}$-AUG$(\mathcal{D})$ and any $F \in \mathbf{F}$. For every non-target $Y \in Adj_{\mathcal{M}}(F) \setminus (tar(F) \cup \{F\})$, adjacency of $F$ and $Y$ in $\mathcal{M} = $MAG$(\mathcal{A})$ implies the existence of an inducing path between $F$ and $Y$ in $\mathcal{A}$. Let $\mathbf{X} \subseteq tar(F)$ be the neighbors of $F$ on such paths or equivalently, a transit set for $(F, Y)$. Collect all such pairs $(X, Y), X \in \mathbf{X}$ across all $F$ into a set $T$ of transit pairs.

Now construct an ADMG $\mathcal{D}'$ on $\mathbf{V}$ by starting from the induced mixed graph $\mathcal{M}[\mathbf{V}]$ and adding bidirected edges $X \leftrightarrow Y$ for every $(X, Y) \in T$. Lemma 5.3 states that adding exactly these bidirected edges fixes the $F$-adjacencies and does not create extra inducing paths among non-adjacent nodes in $\mathcal{M}[\mathbf{V}]$), hence $\mathcal{I}$-MAG$(\mathcal{D}') = \mathcal{M}$. Therefore such a transit pair set $T$ exists.

($\Leftarrow$) Conversely, suppose there exists a set of transit pairs $T$ such that if we define $\mathcal{M}' := \mathcal{M}[\mathbf{V}] \cup \{X \leftrightarrow Y : (X, Y) \in T\}$, then $\mathcal{I}\text{-MAG}(\mathcal{M}') = \mathcal{M}$. Let $\mathcal{D} := \mathcal{M}'$ be an ADMG on $\mathbf{V}$. By assumption, $\mathcal{I}\text{-MAG}(\mathcal{D}) = \mathcal{M}$. Thus $\mathcal{M}$ is realizable as an $\mathcal{I}\text{-MAG}$ and hence is a valid $\mathcal{I}\text{-MAG}$.

Combining both directions proves the equivalence. $\qquad\square$

### B.4. Proof for Theorem 5.7

*Proof.* ($\Rightarrow$) Suppose Algorithm 1 returns $(\texttt{true}, \mathcal{D}, \tau)$. By construction, the procedure verifies at termination that $\text{AUGMAG}(\mathcal{D}, \mathcal{I}) = \mathcal{M}$, where $\texttt{tar}$ is induced by $\mathcal{I}$. Therefore $\mathcal{M} = \mathcal{I}\text{-MAG}(\mathcal{D})$, so $\mathcal{M}$ is a valid $\mathcal{I}\text{-MAG}$.

($\Leftarrow$) Suppose $\mathcal{M}$ is a valid $\mathcal{I}\text{-MAG}$ for $\mathcal{I}$. Then there exists an ADMG $\mathcal{D}^\star$ such that $\mathcal{I}\text{-MAG}(\mathcal{D}^\star) = \mathcal{M}$. By Lemma 5.3, there exists another ADMG $\mathcal{D}'$ constructed from $\mathcal{M}[\mathbf{V}]$ plus bidirected edges for the transit pairs present in $\mathcal{I}\text{-AUG}(\mathcal{D}^\star)$, such that $\mathcal{I}\text{-MAG}(\mathcal{D}') = \mathcal{M}$.

**Corollary B.2** (Single-witness suffices). *Let $\mathcal{M}$ be the output PAG, and let $\mathcal{M}[\mathbf{V}]$ denote the induced subgraph on observational variables $\mathbf{V}$. Let $\mathcal{D}_{\text{all}}$ be the ADMG constructed from $\mathcal{M}[V]$ as in Lemma 5.3 by, for every pair $(F, Y) \in U$, adding a bidirected edge $X \leftrightarrow Y$ for every transit target $X \in Tr(F, Y)$. Let $S$ be any transit selection set such that for every $(F, Y) \in U$ there exists at least one $X \in Tr(F, Y)$ with $(X, F, Y) \in S$, and define $\mathcal{D}_S$ to be the ADMG obtained from $\mathcal{M}[\mathbf{V}]$ by adding only the bidirected edges $X \leftrightarrow Y$ for triples $(X, F, Y) \in S$. Then*

$$\mathcal{I}\text{-MAG}(\mathcal{D}_S) \;=\; \mathcal{I}\text{-MAG}(\mathcal{D}_{\text{all}}) \;=\; \mathcal{M}.$$

*Proof.* First note that $\mathcal{D}_S \subseteq \mathcal{D}_{\text{all}}$ as edge sets, hence $\mathcal{I}\text{-AUG}(\mathcal{D}_S) \subseteq \mathcal{I}\text{-AUG}(\mathcal{D}_{\text{all}})$. Therefore any inducing path present in $\mathcal{I}\text{-AUG}(\mathcal{D}_S)$ is also present in $\mathcal{I}\text{-AUG}(\mathcal{D}_{\text{all}})$, which implies that $\mathcal{I}\text{-MAG}(\mathcal{D}_S)$ cannot contain an adjacency that is absent from $\mathcal{I}\text{-MAG}(\mathcal{D}_{\text{all}}) = \mathcal{M}$.

It remains to show that every adjacency in $\mathcal{M}$ is also present in $\mathcal{I}\text{-MAG}(\mathcal{D}_S)$. All adjacencies within $\mathbf{V}$ are preserved because $\mathcal{M}[\mathbf{V}] \subseteq \mathcal{D}_S$. For any $(F, Y) \in U$, by definition of $S$ there exists $X \in Tr(F, Y)$ with $(X, F, Y) \in S$, so $\mathcal{D}_S$ contains the bow $X \leftrightarrow Y$ and, as in Lemma 5.3, we also have $F \to X$. By the definition of transit nodes and the construction of $\mathcal{I}\text{-AUG}(\cdot)$, the path $F \to X \leftrightarrow Y$ is an inducing path between $F$ and $Y$ in $\mathcal{I}\text{-AUG}(\mathcal{D}_S)$, hence $F$ and $Y$ are adjacent in $\mathcal{I}\text{-MAG}(\mathcal{D}_S)$.

Thus $\mathcal{I}\text{-MAG}(\mathcal{D}_S)$ contains all adjacencies of $\mathcal{M}$ and no additional ones, so $\mathcal{I}\text{-MAG}(\mathcal{D}_S) = \mathcal{M}$. $\qquad\square$

Corollary B.2 claims that for each $(F, Y)$ pair, we just need to identify one transit node instead of the full transit set to recover the $\mathcal{I}\text{-MAG}$. Consider Algorithm 1 executed on input $\mathcal{M}$. For each pair $(F, Y)$ with $Y \in Adj_{\mathcal{M}}(F) \setminus (tar(F) \cup \{F\})$, the transit node used in $\mathcal{D}'$ is adjacent to $Y$ in $\mathcal{M}[\mathbf{V}]$, hence it belongs to the candidate set $Cand(F, Y)$ constructed by the algorithm. Therefore the depth-first search performed by Algorithm 6 contains a branch that selects a transit node which is contained in the true transit set. Along this branch, the pruning test $\text{MAG}(\mathcal{D}') \equiv \mathcal{M}[\mathbf{V}]$ never rejects, because by Corollary B.2, the construction of $\mathcal{D}'$ by adding a bidirected edge between a transit node and a non-target node only fixes the adjacency of corresponding $F$ nodes while not creating inducing paths between non-adjacent nodes in $\mathcal{D}'$. Hence the recursion eventually reaches a leaf where $\text{AUGMAG}(\mathcal{D}', \mathcal{I}) = \mathcal{M}$, and Algorithm 1 returns success.

Therefore Algorithm 1 returns a non-null output. $\qquad\square$

### B.5. Proof for Theorem 5.8

*Proof.* Let $\mathcal{P}_0$ denote the output of $\mathcal{I}\text{-FCI}$ but without applying $\mathcal{I}\text{-FCI}$'s Rule 9. It has been shown that the $\mathcal{P}_0$ is a sound structure in Kocaoglu et al. (2019). Therefore, we just need show that the learning process after $\mathcal{P}_0$ is sound and complete.

After getting $\mathcal{P}_0$, Algorithm 2 calls the MAG listing Algorithm in Wang et al. (2025) to enumerate all valid MAGs with the same configurations as $\mathcal{P}_0$. The MAG listing algorithm takes PAG as input. Although $\mathcal{P}_0$ is not an observable PAG by definition, it is equivalent as an observable PAG with local BK at all $F$ nodes. To witness, $\mathcal{P}_0$ contains all $\mathcal{I}\text{-MAGs}$ constructed with edges outgoing from $F$ nodes. Further, it is then fully oriented with FCI rules. Hence, $\mathcal{P}_0$ has all the graphical properties of observational PAGs. Now that the orientation around $F$ are valid by construction, $\mathcal{P}_0$ can be viewed as a branch of running the MAG listing algorithm on the skeleton over $\mathbf{V} \cup \mathbf{F}$ that already orients edges outgoing from $F$ nodes. Hence, MAGLIST-POLY will not reject $\mathcal{P}_0$ and list any MAG that is entailed by $\mathcal{P}_0$. All valid $\mathcal{I}\text{-MAGs}$ will be

included in the listed MAGs since $\mathcal{I}$-MAGs are MAGs by their construction. Now that Algorithm 1 will recognize any valid $\mathcal{I}$-MAG from the listed MAGs, we will not miss any valid $\mathcal{I}$-MAG. Taking the edge marks of the union graph at each step, we get the $\mathcal{I}$-PAG as $\widehat{\mathcal{P}}$ by definition. Therefore, Algorithm 2 is sound and complete. □

### B.6. Proof for Theorem 5.9

*Proof.* We argue each rule separately. Consider an $\mathcal{I}$-MEC and let $\mathcal{M}$ be any valid $\mathcal{I}$-MAG in it. Let $\mathcal{P}$ denote the partially oriented graph obtained after applying Rule 0, Zhang's FCI rules, and orienting all edges out of $F$ nodes. Standard soundness of the FCI closure implies: any arrowhead already present in $\mathcal{P}$ is invariant across all $\mathcal{I}$-MAGs consistent with the $\mathcal{I}$-MEC .

**Rule 9.** Let $Y$ be $F$-adjacent and non-target, and assume that among the targets adjacent to $Y$, there is exactly one target $X \in tar(F)$ that is a *possible parent* of $Y$ in $\mathcal{P}$. Equivalently, for every other target $X' \in tar(F) \cap Adj_{\mathcal{P}}(Y)$, the endpoint mark at $X'$ on the edge $X' *\!\!-\!* Y$ is an arrowhead, hence $X' \to Y$ is forbidden.

Because $F$ and $Y$ are adjacent in $\mathcal{M}$, Lemma 5.1 implies that there exists a transit target $X^\star \in tar(F)$ witnessing this adjacency, and moreover $X^\star \in An_{\mathcal{D}}(Y)$. Any transit node must be adjacent to $Y$ in the induced MAG on $\mathbf{V}$ (otherwise no inducing witness segment from $X^\star$ to $Y$ could exist), hence $X^\star \in tar(F) \cap Adj_{\mathcal{M}}(Y)$. Since $\mathcal{P}$ is a sound partial representation of the $\mathcal{I}$-MEC , every arrowhead at a target endpoint in $\mathcal{P}$ forbids that target from being a parent of $Y$ in *any* consistent MAG. Therefore $X^\star$ must lie in the set of possible parents of $Y$ in $\mathcal{P}$ and the targets of this $F$ node, which is a singleton $\{X\}$. Hence $X^\star = X$.

Finally, since $X \in An_{\mathcal{D}}(Y)$ and $X$ is adjacent to $Y$ in an ancestral graph, the edge between $X$ and $Y$ cannot have an arrowhead at $X$ in any consistent MAG. Under no-selection-bias assumptions, this forces $X \to Y$ in every consistent $\mathcal{I}$-MAG. Therefore orienting $X \to Y$ is sound, and recording $(X, Y)$ as a transit pair is correct.

**Rule 10.** Assume Rule 10 applies to nodes $(Z, X, Y)$ where $(X, Y)$ has been recorded as a transit pair. By Lemma 5.1, there exists an inducing path from $F$ to $Y$ in the augmented graph whose neighbor of $F$ is $X$. Equivalently, there exists a witness segment from $X$ to $Y$ that begins with an arrowhead into $X$, and all colliders on the segment are ancestors of $Y$. Suppose for contradiction that the edge between $Z$ and $X$ has an arrowhead into $X$ in some consistent MAG, i.e., $Z*\!\!\to X$. Concatenating the edge $Z*\!\!\to X$ with the witness segment from $X$ to $Y$ yields an inducing path between $Z$ and $Y$, since $X$ becomes a collider with two arrowheads into it, and all other colliders remain ancestors of $Y$. This would imply that $Z$ and $Y$ are adjacent in the MAG, contradicting the premise of Rule 10 that $Z$ is not adjacent to $Y$. Hence no arrowhead into $X$ is possible on $Z\!-\!X$, and ancestrality forces $X \to Z$, i.e., $Z \leftarrow X$. Thus Rule 10 is sound.

**Rule 11.** Let $Y$ be $F$-adjacent and non-target, and suppose $Y$ is adjacent to multiple targets in $tar(F)$. Assume further that among these target neighbors, $X$ is the unique *source* in the directed target-induced subgraph, i.e., $X$ has no incoming directed edge from another target in this set, and every other target has at least one incoming directed edge from within the set.

By Lemma 5.1, at least one target neighbor $T$ of $Y$ is a transit target for $(F, Y)$, hence $T \in An_{\mathcal{D}}(Y)$. If $T = X$, then $X \in An\mathcal{D}(Y)$ and adjacency of $X$ and $Y$ forces $X \to Y$ as in Rule 9.

Otherwise $T \neq X$. Since every non-source target has an incoming directed edge from within the target set, following directed in-edges backward from $T$ must terminate at a source node in this finite acyclic directed subgraph. By uniqueness of the source, this terminal node is $X$, which yields a directed path $X \to \cdots \to T$. Therefore $X \in An_{\mathcal{D}}(T)$. Since $T \in An_{\mathcal{D}}(Y)$, transitivity implies $X \in An_{\mathcal{D}}(Y)$. Because $X$ is adjacent to $Y$ in the MAG, ancestrality again forces the edge to be $X \to Y$ in every consistent $\mathcal{I}$-MAG. Hence Rule 11 is sound.

Since each rule only orients endpoint marks that hold in every consistent $\mathcal{I}$-MAG, the three rules are sound. □

## C. Detailed Algorithms

---

**Algorithm 3** BASICFCHECKS

---

> **Input:** $\mathcal{M}$ on $\mathbf{V} \cup \mathbf{F}, \mathcal{I}$
> **Output:** true/false
> **for** $F \in \mathbf{F}$ **do**
>   **if** $F$ has any incident arrowhead in $\mathcal{M}$ **then**
>     **return** false
>   **end if**
>   **for** $X \in tar(F)$ **do**
>     **if** $F \to X$ is not in $M$ **then**
>       **return** false
>     **end if**
>   **end for**
> **end for**
> **return** true

---

**Algorithm 4** BUILDCANDIDATES

---

> **Input:** $\mathcal{M}, \mathcal{D}_0 = \mathcal{M}[\mathbf{V}], \mathcal{I}, \mathbf{P}$
> **Output:** candidate map $Cand$ and ordered list $\mathbf{P}'$, or fail
> **for** $(F, Y) \in \mathbf{P}$ **do**
>   $Cand(F, Y) \leftarrow \{X \in tar(F) : X \in Pa_{\mathcal{D}_0}(Y)\}$
>   **if** $Cand(F, Y) = \emptyset$ **then**
>     **return** fail
>   **end if**
> **end for**
> Order $\mathbf{P}$ by increasing $|Cand(F, Y)|$ to obtain $\mathbf{P}'$
> **return** $(Cand, \mathbf{P}')$

---

**Algorithm 5** AUGMAG

---

> **Input:** ADMG $\mathcal{D}$ on $\mathbf{V}$, intervention targets $\mathcal{I}$
> **Output:** mixed graph $\mathcal{M}$ (the $\mathcal{I}$-MAG on $\mathbf{V} \cup \mathbf{F}$)
> Initialize an augmented mixed graph $\mathcal{A} \leftarrow \mathcal{D}$
> Construct the set of $F$-nodes for each unordered pair $I, J \in \mathcal{I}$
>   $\mathbf{F} \leftarrow \{F_{\{I,J\}} : I, J \in \mathcal{I}, \ I \neq J\}$
> Add vertices $\mathbf{F}$ to $\mathcal{A}$
> **for** each $F_{\{I,J\}} \in \mathbf{F}$ **do**
>   $tar(F_{\{I,J\}}) \leftarrow I \Delta J$
>   **for** each $X \in tar(F_{\{I,J\}})$ **do**
>     add $F_{\{I,J\}} \to X$ to $\mathcal{A}$ if absent
>   **end for**
> **end for**
> $\mathcal{M} \leftarrow \text{MAG}(\mathcal{A})$
> **return** $\mathcal{M}$

---

---

**Algorithm 6** SEARCHTRANSIT

---

**Input:** A mixed graph $\mathcal{M}$, base ADMG $\mathcal{D}_0$, $\mathcal{I}$, $Cand$, ordered constraints $\mathbf{P}'$
**Output:** $(ok, \tau, \mathcal{D})$
Initialize $\tau \leftarrow \emptyset$, $\mathcal{D} \leftarrow \mathcal{D}_0$
$(ok, \tau, \mathcal{D}) \leftarrow \text{DFS}(1, \tau, \mathcal{D})$
**return** $(ok, \tau, \mathcal{D})$
**procedure** DFS$(i, \tau, \mathcal{D})$
**if** $i > |\mathbf{P}'|$ **then**
  **if** AUGMAG$(\mathcal{D}, \mathcal{I}) = \mathcal{M}$ **then**
    **return** $(\texttt{true}, \tau, \mathcal{D})$
  **else**
    **return** $(\texttt{false}, \emptyset, \emptyset)$
  **end if**
**end if**
$(F, Y) \leftarrow \mathbf{P}'[i]$
**for** $X \in Cand(F, Y)$ **do**
  $\mathcal{D}' \leftarrow \mathcal{D}$;   add $X \leftrightarrow Y$ to $\mathcal{D}'$ if absent
  $\tau' \leftarrow \tau \cup \{(F, Y) \mapsto X\}$
  **if** MAG$(\mathcal{D}')$ is undefined **or** MAG$(\mathcal{D}') \not\equiv \mathcal{M}[\mathbf{V}]$ **then**
    **continue**
  **end if**
  $(ok, \tau^\star, \mathcal{D}^\star) \leftarrow \text{DFS}(i+1, \tau', \mathcal{D}')$
  **if** $ok$ **then**
    **return** $(\texttt{true}, \tau^\star, \mathcal{D}^\star)$
  **end if**
**end for**
**return** $(\texttt{false}, \emptyset, \emptyset)$
**end procedure**

---

**Algorithm 7** INTERSECTMARKS

---

**Input:** A fully oriented mixed graph $\mathcal{M}$, edge mark for each endpoint $Mark$
**Output:** Updated edge marks $Mark$
**for** each endpoint $e$ in $Mark$ **do**
  **if** $Mark(e) = \emptyset$ **then**
    Set $Mark(e)$ to edge mark of $e$ in $\mathcal{M}$
  **else if** $Mark(e)$ is the same as edge mark of $e$ in $\mathcal{M}$ **OR** $Mark(e)$ is circle **then**
    Pass
  **else if** $Mark(e)$ is arrowhead, edge mark of $e$ is arrowtail **OR** $Mark(e)$ is arrowtail, edge mark of $e$ is arrowhead
  **then**
    Set $Mark(e)$ to circle
  **end if**
**end for**
**return** $Mark$

---

---

**Algorithm 8** $\mathcal{I}$-FCI-BASE: Build Augmented Graphs and Learn $\mathcal{P}_0$

---

**Input:** intervention targets $\mathcal{I}$, interventional distributions $(P_I)_{I \in \mathcal{I}}$, observed variables $\mathbf{V}$
**Output:** partially oriented graph $\mathcal{P}_0$ on $\mathbf{V} \cup \mathbf{F}$ and the set of $F$-nodes $\mathbf{F}$
**Phase I: Create $F$ nodes and initialize a complete graph**
$\mathbf{F} \leftarrow \emptyset$;    $\mathcal{P}_0 \leftarrow$ complete circle graph on $\mathbf{V}$ (i.e., $X \circ\!\!-\!\!\circ Y$ for all $X \neq Y \in \mathbf{V}$)
**for** each unordered pair $\{I, J\}$ with $I, J \in \mathcal{I}, I \neq J$ **do**
    Create a new intervention node $F_{\{I,J\}}$
    $\mathbf{F} \leftarrow \mathbf{F} \cup \{F_{\{I,J\}}\}$
    Add $F_{\{I,J\}}$ to $\mathcal{P}_0$ and connect it to every $V \in \mathbf{V}$ with a circle edge $F_{\{I,J\}} \circ\!\!-\!\!\circ V$
**end for**
**Phase II: Skeleton learning and separating sets**
Initialize $SepSet \leftarrow \emptyset$
Run the standard $\mathcal{I}$-FCI skeleton learning procedure of Kocaoglu et al. (2019) on $(P_I)_{I \in \mathcal{I}}$, with node set $\mathbf{V} \cup \mathbf{F}$, to:
    (i) delete edges in $\mathcal{P}_0$ using CI tests and do-constraint tests; and
    (ii) store separating sets $SepSet(X, Z)$ for non-adjacent pairs $(X, Z)$.
**Phase III: Initial orientations (colliders and $F$-out edges)**
**for** each unshielded triple $\langle X, Y, Z \rangle$ in $\mathcal{P}_0$ with $X$ not adjacent to $Z$ **do**
    **if** $Y \notin SepSet(X, Z)$ **then**
        Orient $X \ast\!\!-\!\!\circ Y \circ\!\!-\!\!\ast Z$ as $X \rightarrow Y \leftarrow Z$
    **end if**
**end for**
**for** each $F \in \mathbf{F}$ and each neighbor $V \in Adj_{\mathcal{P}_0}(F)$ **do**
    Orient $F \circ\!\!-\!\!\ast V$ as $F \rightarrow V$
**end for**
Apply the 7 FCI rules to $\mathcal{P}_0$ until closure
**return** $(\mathcal{P}_0, \mathbf{F})$

---

# D. Time Complexity Analysis

This section provides a high-level time-complexity discussion for the two auxiliary procedures introduced in the main text, the $\mathcal{I}$-MAG realizability oracle (Algorithm 1) and the completion procedure based on MAG listing plus $\mathcal{I}$-MAG filtering (Algorithm 2). We emphasize that the running times are *output-sensitive* in the sense typical for equivalence-class enumeration.

**Notation.** Let $d := |\mathbf{V}|$ be the number of observed variables and let $m := |E(\mathcal{M}[\mathbf{V}])|$ be the number of edges in the induced mixed graph on $\mathbf{V}$. Let $\mathbf{F}$ be the set of $F$-nodes. For each $F \in \mathbf{F}$, let $tar(F) \subseteq \mathbf{V}$ denote its target set. Define the number of non-target $F$-adjacencies

$$r := \sum_{F \in \mathbf{F}} \Big| Adj_{\mathcal{M}}(F) \setminus (tar(F) \cup \{F\}) \Big|,$$

i.e., the total number of pairs $(F, Y)$ for which Algorithm 1 must select at least one transit target. When discussing Algorithm 2, let $N$ denote the number of MAGs consistent with the partially oriented input graph $\mathcal{P}_0$ produced by $\mathcal{I}$-FCI .

## D.1. Algorithm 1: $\mathcal{I}$-MAG realizability

Let $s := |\mathcal{I}|$ be the number of domains and let $k := \max_{I \in \mathcal{I}} |I|$ be the maximum intervention target size. Let $\mathbf{F}$ be the set of $F$-nodes (one per unordered pair of intervention targets), hence $|\mathbf{F}| = O(s^2)$.

Algorithm 1 introduces one constraint for each pair $(F, Y)$ where $Y$ is a non-target adjacent to $F$ in $\mathcal{M}$. As mentioned, $r$ denotes the total number of such constraints. For each $(F, Y)$, the candidate set $Cand(F, Y)$ has size at most $2k$ since $Cand(F, Y) \subseteq tar(F)$. Algorithm 1 performs a depth-first search over candidate choices and uses a polynomial-time feasibility test per node for checking that $\mathrm{MAG}(\mathcal{D}') \equiv \mathcal{M}[\mathbf{V}]$. Concretely, at each DFS node we form an ADMG $\mathcal{D}'$ on $\mathbf{V}$ and compute a corresponding MAG representative $\mathrm{MAG}(\mathcal{D}')$. Computing the maximalization of an ADMG to an equivalent MAG can be done in polynomial time, see, e.g., Hu & Evans (2020) who give an $O(d^2 m)$ algorithm. More generally, the existence of a polynomial-time maximalization procedure for ancestral graphs is noted by Ali et al. (2009). Finally, once $\mathrm{MAG}(\mathcal{D}')$ is obtained, checking $\mathrm{MAG}(\mathcal{D}') = \mathcal{M}[\mathbf{V}]$ takes $O(m)$ time.

Consequently, the running time is

$$T_1(d, m) = O\big((2k)^r \cdot \mathrm{poly}(d, m)\big),$$

i.e., fixed-parameter tractable in $(k, r)$.

*Polynomial-time regime.* In many applications, both the intervention target size $k$ and the number of $F$-adjacent non-targets per $F$ are small. If we additionally assume

$$\max_{F \in \mathbf{F}} \Big| Adj_{\mathcal{M}}(F) \setminus (tar(F) \cup \{F\}) \Big| \leq b$$

for a constant $b$, then $r \leq O(s^2 b)$ is constant when $s$ is also treated as constant, Algorithm 1 runs in polynomial time in $d$ and $m$.

## D.2. Algorithm 2: Enumeration-based Completion

Algorithm 2 first runs $\mathcal{I}$-FCI -BASE to obtain a partially oriented graph $\mathcal{P}_0$ on the vertex set $\mathbf{V} \cup \mathbf{F}$. It then enumerates all MAG completions consistent with $\mathcal{P}_0$ using MAGLIST-POLY by Wang et al. (2025), filters each candidate MAG via the realizability oracle in Algorithm 1, and finally intersects endpoint marks across all accepted candidates.

Let $n := |\mathbf{V} \cup \mathbf{F}| = d + |\mathbf{F}|$ be the total number of vertices in $\mathcal{P}_0$, and let $m_0$ be the number of edges in the skeleton of $\mathcal{P}_0$. Let $N$ denote the number of MAGs consistent with $\mathcal{P}_0$ (i.e., the output size of the listing procedure). For a listed MAG $\mathcal{M}$, recall that Algorithm 1 only searches over edges on the observed subgraph $\mathcal{M}[V]$.

**Per-candidate overhead.** The MAG listing algorithm of Wang et al. (2025) provides polynomial delay for enumerating the $N$ MAG completions. Let $Delay(n, m_0)$ denote its worst-case delay bound expressed in terms of $n$ and $m_0$. For each listed candidate $\mathcal{M}$, Algorithm 2 additionally runs the realizability oracle with cost $T_1$ as derived in Section D.1.

**Total running time.** Thus, Algorithm 2 is output-sensitive in the number of listed MAGs:

$$T_2 = O\left(\sum_{i=1}^{N}\left[Delay(n, m_0) + T_1\big(d, m(\mathcal{M}_i), r(\mathcal{M}_i)\big)\right]\right),$$

where $\{\mathcal{M}_1, \ldots, \mathcal{M}_N\}$ are the MAGs enumerated by MAGLIST-POLY. In particular, letting $m_{\max} := \max_i m(\mathcal{M}_i)$ and $r_{\max} := \max_i r(\mathcal{M}_i)$ yields the coarse bound

$$T_2 = O\left(N \cdot Delay(n, m_0) + N \cdot (2k)^{r_{\max}} poly(d, m_{\max})\right).$$

The final mark-intersection step INTERSECTMARKS costs $O(m_0)$ per accepted candidate and is dominated by the listing and realizability costs.

**Worst-case cost.** Although Algorithm 2 is output-sensitive and the listing subroutine has polynomial delay, the overall runtime is dominated by the number $N$ of MAG completions consistent with $\mathcal{P}_0$. In the worst case, $N$ can be *super-exponential* in the number of vertices $n = |V \cup F|$ (i.e., it may grow faster than $c^n$ for any fixed constant $c > 1$), so $T_2$ is not polynomially bounded in general. Moreover, because Algorithm 1 filters listed MAGs using the realizability oracle, the time between two *accepted* $\mathcal{I}$-MAG can be much larger than the polynomial-delay bound for listing, if many candidates are rejected. Consequently, Algorithm 2 should be viewed primarily as a *theoretical completion* procedure rather than a uniformly efficient end-to-end learner.

**Remark (witness-based identifiability checks).** After obtaining $\mathcal{P}_0$, our goal is to determine which remaining circle endpoints are in fact identifiable under the $\mathcal{I}$-MAG semantics. This can be formulated as a collection of existence queries rather than a full enumeration task. Specifically, for each circle endpoint $c$ in $\mathcal{P}_0$, we consider the two possible assignments of $c$, arrowtail vs. arrowhead, and ask whether there exists at least one valid $\mathcal{I}$-MAG completion of $\mathcal{P}_0$ that realizes the assignment. If a witness $\mathcal{I}$-MAG exists for exactly one of the two assignments, then $c$ is identifiable and we can orient it accordingly. If witnesses exist for both assignments, then $c$ is not identifiable and must remain a circle. A practical implementation is to modify the MAG listing procedure into a backtracking search that is biased to realize the queried mark and terminates as soon as a witness passing the realizability test in Algorithm 1 is found. This approach can yield substantial empirical speedups because many queries terminate after finding a single witness, while still being sound and complete when the underlying search is exhaustive, i.e., it proves non-existence by exploring all completions consistent with the constraints. We do not analyze this variant further since it is orthogonal to the main conceptual contributions of the paper.

### D.3. Complexity of the Local Orientation Rules (Rules 9–11)

We briefly discuss the computational cost of applying the three additional local orientation rules. Let $\Delta$ denote the maximum degree in $\mathcal{P}$. Let $s := |\mathcal{I}|$ be the number of domains and $|\mathbf{F}| = O(s^2)$ the number of $F$ nodes.

In our implementation, Rules 9–11 are applied repeatedly until no rule is applicable, interleaved with the standard FCI closure rules. We analyze the worst-case cost of one full pass over all vertices and edges. The number of passes is polynomially bounded because each application strictly reduces the number of circle marks or orients at least one endpoint, and there are at most $2m$ endpoint marks.

**Precomputation.** We maintain adjacency lists and, for each vertex $Y$, the set of $F$-neighbors $Adj_{\mathbf{F}}(Y) := Adj_{\mathcal{P}}(Y) \cap \mathbf{F}$ and target-neighbors $Adj_{tar(F)}(Y) := Adj_{\mathcal{P}}(Y) \cap tar(F)$ for each $F$. These can be updated incrementally after each orientation and cost $O(\Delta)$ per updated endpoint. Over all endpoints this yields $O(m\Delta)$ total update cost across the entire run.

**Rule 9.** Rule 9 triggers for a pair $(F, Y)$ where $Y$ is non-target and adjacent to $F$, and among target neighbors of $Y$ there is exactly one target $X \in tar(F)$ that remains a possible parent of $Y$ in $\mathcal{P}$. For a fixed $Y$, computing the set

$$T_F(Y) := \{X \in tar(F) : X \in Adj_{\mathcal{P}}(Y) \text{ and the mark at } X \text{ on } X \ast\!\!-\!\!\ast Y \text{ is not an arrowhead}\}$$

takes $O(\deg(Y))$ time by scanning neighbors and checking endpoint marks. Summed over all $Y$, a full pass costs $O(m)$ time. We can bound the target size by $2k$, then scanning can be restricted to $Adj_{\mathcal{P}}(Y) \cap tar(F)$, yielding an $O(rk)$ total time complexity per pass, where $r$ is the number of $F$-to-non-target adjacency constraints.

**Rule 10.** Rule 10 is applied when a transit pair $(X, Y)$ has been identified and for any $Z \in Adj_P(X) \setminus Adj_{\mathcal{P}}(Y)$ the endpoint mark at $X$ on $Z \ast\!\!-\!\!\ast X$ is oriented as a tail, equivalently, forbid $Z \ast\!\!\rightarrow X$. For a fixed transit pair $(X, Y)$, computing $Adj_{\mathcal{P}}(X) \setminus Adj_{\mathcal{P}}(Y)$ costs $O(\deg(X) + \deg(Y))$ using adjacency hashing, and then orienting the relevant endpoint marks costs $O(\deg(X))$. Hence one application costs $O(\deg(X) + \deg(Y)) \leq O(\Delta)$, and over all possible transit pairs the total cost is $O(m\Delta)$ in the worst case. With incremental maintenance of neighbor sets, this becomes $O(\deg(X))$ amortized per triggered transit pair.

**Rule 11.** Rule 11 triggers for a pair $(F, Y)$ where $Y$ is non-target and adjacent to $F$, and among the target neighbors of $Y$ there is a unique source in the *directed* subgraph induced by those targets. For a fixed $(F, Y)$, let $T := Adj_{\mathcal{P}}(Y) \cap tar(F)$ and consider the directed edges among $T$. Computing in-degrees in this induced directed subgraph can be done in $O(|T|^2)$ time by checking all pairs, or in $O(|E(T)|)$ time by scanning edges incident to vertices in $T$ and counting directed in-edges. Since $|T| \leq k$ as it is bounded by target size, this is $O(k^2)$ per $(F, Y)$ in the worst case. Therefore a full pass over all $(F, Y)$ constraints costs $O(rk^2)$, which is $O(r)$ when $k$ is treated as a constant.

**Overall cost.** Let $C$ be the total number of rule applications with each application orients at least one endpoint mark, so $C \leq 2m$. Using the bounds above and maintaining adjacency information incrementally, the total running time for applying Rules 9–11 to convergence is polynomial:

$$T_{\text{Rules}} = O(m\Delta + rk^2) \cdot O(1) \text{ passes} \leq \text{poly}(d, m),$$

and under bounded target size ($k = O(1)$) this simplifies to $T_{\text{Rules}} = O(m\Delta + r)$. In particular, the additional local rules do not change the asymptotic complexity class of the baseline FCI-style closure procedure, and add at most a low-order polynomial overhead per iteration.

# E. Experiment Details and Runtime Discussion

For the main synthetic experiments, we randomly generate ADMGs following the same procedure as in Section 6. We first sample a random topological order over the observed nodes and generate directed edges with density $\rho_{\text{DAG}} = 0.5$. We then independently add bidirected edges according to the latent-confounding density $\rho_{\text{bi}}$. Given each generated ADMG, we construct a binary Bayesian network using pyAgrum (Ducamp et al., 2020), with randomly sampled conditional probability tables. For each domain, we sample $50,000$ observations. We use three domains, one of which is observational, and each intervention target has maximum size 2. If a variable appears in multiple intervention targets, we impose the same shifted mechanism for that variable across those domains, matching the controlled soft-intervention assumption.

For evaluation, we compare the learned augmented graph against the ground-truth augmented MAG. For each sampled ADMG, we first attach the $F$-nodes according to the symmetric differences of the intervention targets and then compute the MAG of the resulting augmented graph. The main text reports PG-SHD and End-F1 on the common successful trials across all compared methods.

The three methods have similar runtime in the main $n = 5$ experiments. Averaged over the common successful trials, the runtimes for $\rho_{\text{bi}} = 0.5$ are $7.42 \pm 1.61$ seconds for $\mathcal{I}$-FCI , $7.44 \pm 1.61$ seconds for FAST, and $7.54 \pm 1.69$ seconds for LIST. For $\rho_{\text{bi}} = 0.8$, the corresponding runtimes are $7.58 \pm 1.85$, $7.60 \pm 1.85$, and $7.77 \pm 1.87$ seconds. Thus, FAST has essentially the same runtime as $\mathcal{I}$-FCI in these experiments, supporting that the additional local rules add only a small overhead.

Although LIST is theoretically complete under exact inputs, it can be less robust under finite-sample errors. In finite samples, the skeleton learned during the initial $\mathcal{I}$-FCI phase may contain missing or extra adjacencies. Since LIST enforces global $\mathcal{I}$-MAG realizability, such skeleton errors can make the partially oriented graph incompatible with any valid $\mathcal{I}$-MAG completion, causing the realizability filter to reject all candidates. FAST, in contrast, only applies sound local rules when their graphical preconditions are met, and therefore can still return a refinement of the learned partial graph even when the global realizability constraints are not satisfiable. This explains why LIST may fail on some trials despite being complete in the population-level setting.

# F. Additional Experiments

We provide additional experiments to evaluate the behavior of $\mathcal{I}$-FCI , FAST, and LIST beyond the small synthetic setting in the main text. We report skeleton structural Hamming distance (Skel-SHD), partial-graph structural Hamming distance

(PG-SHD), and wall-clock runtime in seconds. Skel-SHD counts adjacency errors, while PG-SHD additionally penalizes endpoint-mark disagreements. Lower values are better for both metrics. Unless otherwise stated, results are reported as mean $\pm$ standard deviation over independent runs.

## F.1. Real-World Evaluation on the Sachs Dataset

We evaluate the methods on the Sachs protein-signaling dataset (Sachs et al., 2005). In each run, we randomly select two experimental domains and treat the variables whose mechanisms change across the selected domains as intervention targets. When the perturbed node is not directly observed, we use the closest downstream observed node as the target. Since the reference graph derived from the literature is not a true ground-truth graph, this experiment should be interpreted as a qualitative robustness check rather than a definitive accuracy benchmark.

*Table 2.* Performance on the Sachs dataset. PG-SHD is computed against the literature-derived reference graph. Lower is better.

| Method | PG-SHD $\downarrow$ | Runtime (s) $\downarrow$ |
|---|---|---|
| $\mathcal{I}$-FCI | $20.00 \pm 0.55$ | $0.0897 \pm 0.0065$ |
| FAST | $19.80 \pm 0.45$ | $0.0994 \pm 0.0167$ |
| LIST | $19.20 \pm 0.84$ | $8.1648 \pm 0.6423$ |

FAST and LIST slightly improve over $\mathcal{I}$-FCI in PG-SHD, while LIST is substantially slower. The modest improvement may be due to weak latent-confounding effects in this dataset, in which case the additional $F$-node orientation principles do not frequently trigger.

## F.2. Scaling to 100 Observable Nodes

We next evaluate scalability on larger random ADMGs with 100 observable nodes. In each run, we generate a sparse ADMG, randomly select two intervention targets with maximum target size 7, and compare FAST against $\mathcal{I}$-FCI . LIST is terminated after exceeding 500 seconds on a single run.

*Table 3.* Scaling experiment with 100 observable nodes. LIST is omitted because it exceeded 500 seconds in a single run.

| Method | PG-SHD $\downarrow$ | Runtime (s) $\downarrow$ |
|---|---|---|
| $\mathcal{I}$-FCI | $813.20 \pm 28.59$ | $105.21 \pm 0.13$ |
| FAST | $810.70 \pm 28.63$ | $105.23 \pm 0.17$ |
| LIST | – | $>500$s |

FAST gives a small improvement over $\mathcal{I}$-FCI while incurring essentially no additional runtime. This supports the claim that Rules 9–11 add only low-order computational overhead. The improvement is modest because the new rules trigger only under specific local configurations, which may be rare in sparse random graphs.

## F.3. Effect of Sample Size

We evaluate the effect of sample size on synthetic graphs with $n = 10$ observable nodes. We report both Skel-SHD and PG-SHD to separate adjacency recovery from endpoint-mark recovery.

*Table 4.* Effect of sample size on synthetic graphs with $n = 10$ observable nodes. Lower is better for SHD metrics.

| $N$ | FAST | | | $\mathcal{I}$-FCI | | | LIST | | |
|---|---|---|---|---|---|---|---|---|---|
| | Skel-SHD $\downarrow$ | PG-SHD $\downarrow$ | Runtime (s) $\downarrow$ | Skel-SHD $\downarrow$ | PG-SHD $\downarrow$ | Runtime (s) $\downarrow$ | Skel-SHD $\downarrow$ | PG-SHD $\downarrow$ | Runtime (s) $\downarrow$ |
| 2500 | $14.16 \pm 3.30$ | $23.52 \pm 4.09$ | $0.08 \pm 0.01$ | $14.16 \pm 3.30$ | $23.72 \pm 4.11$ | $0.08 \pm 0.01$ | $14.16 \pm 3.30$ | $23.52 \pm 4.09$ | $22.59 \pm 28.76$ |
| 5000 | $13.04 \pm 3.53$ | $23.58 \pm 5.61$ | $0.14 \pm 0.01$ | $13.04 \pm 3.53$ | $23.82 \pm 5.63$ | $0.14 \pm 0.01$ | $13.04 \pm 3.53$ | $23.66 \pm 5.33$ | $38.38 \pm 40.26$ |
| 10000 | $13.30 \pm 3.19$ | $23.74 \pm 5.05$ | $0.27 \pm 0.01$ | $13.30 \pm 3.19$ | $24.22 \pm 4.82$ | $0.26 \pm 0.01$ | $13.30 \pm 3.19$ | $23.78 \pm 4.64$ | $41.67 \pm 44.93$ |
| 25000 | $12.28 \pm 3.15$ | $24.20 \pm 4.74$ | $0.64 \pm 0.04$ | $12.28 \pm 3.15$ | $24.70 \pm 4.61$ | $0.64 \pm 0.04$ | $12.28 \pm 3.15$ | $24.44 \pm 4.68$ | $68.92 \pm 40.93$ |
| 50000 | $11.94 \pm 3.09$ | $24.76 \pm 5.34$ | $1.25 \pm 0.08$ | $11.94 \pm 3.09$ | $25.08 \pm 5.26$ | $1.25 \pm 0.08$ | $11.94 \pm 3.09$ | $24.38 \pm 4.83$ | $69.65 \pm 41.26$ |

As expected, Skel-SHD generally improves as the sample size increases, reflecting better adjacency recovery from more reliable CI tests. PG-SHD is not monotone because it penalizes endpoint marks and compares a partial graph to a single reference MAG. With more samples, the algorithms may commit to more orientations; these additional endpoint commitments can increase PG-SHD when some directions are not identifiable from the available interventional information

or when finite-sample CI errors propagate through orientation rules. This motivates reporting Skel-SHD and PG-SHD separately.

### F.4. Effect of Maximum Target Size

We also vary the maximum intervention target size in synthetic graphs with $n = 10$ observable nodes and two domains. Each setting is averaged over 50 runs.

*Table 5.* Effect of maximum intervention target size. Lower is better.

| Method | Max target size | PG-SHD $\downarrow$ | Runtime (s) $\downarrow$ |
|---|---|---|---|
| $\mathcal{I}$-FCI | 1 | $24.16 \pm 4.85$ | $0.060 \pm 0.003$ |
| FAST | 1 | $24.14 \pm 4.85$ | $0.061 \pm 0.004$ |
| LIST | 1 | $24.15 \pm 3.30$ | $2.543 \pm 0.879$ |
| $\mathcal{I}$-FCI | 2 | $24.08 \pm 5.19$ | $0.060 \pm 0.002$ |
| FAST | 2 | $23.80 \pm 5.24$ | $0.061 \pm 0.003$ |
| LIST | 2 | $22.75 \pm 3.82$ | $2.469 \pm 0.919$ |
| $\mathcal{I}$-FCI | 3 | $23.58 \pm 5.00$ | $0.060 \pm 0.003$ |
| FAST | 3 | $23.24 \pm 5.12$ | $0.061 \pm 0.003$ |
| LIST | 3 | $22.36 \pm 4.27$ | $2.435 \pm 0.999$ |

The improvements become more visible as the maximum target size increases. This is consistent with the role of known targets in our theory: larger target sets create more opportunities for $F$-adjacencies to construct non-target edges and further identify transit nodes. FAST remains as efficient as $\mathcal{I}$-FCI , while LIST obtains stronger orientation recovery at the cost of substantially higher runtime.

### F.5. Summary

Across these additional experiments, FAST consistently preserves the computational efficiency of $\mathcal{I}$-FCI while recovering slightly more endpoint information. LIST can recover additional marks, especially when the learned skeleton is compatible with a valid $\mathcal{I}$-MAG, but it is much slower and can be brittle under finite-sample CI errors because it enforces global realizability constraints. These results support the interpretation of FAST as an efficient local refinement and LIST as a theoretically complete but computationally expensive completion procedure.

### F.6. Environment

All experiments are implemented in Python 3.11 (Anaconda). Conditional independence tests use `scipy`, data are simulated with `pyAgrum`, and all runs are executed on a machine equipped with an NVIDIA GeForce RTX 3060 Ti GPU.

## G. Further Discussion

### G.1. Why the Local Rules Are Not Complete

The local orientation rules in FAST are sound but not complete. The main reason is that some compelled orientations depend on the global compatibility of transit assignments across multiple non-target edges. Rules 9–11 are designed to use local information around an $F$-node: if a non-target node adjacent to $F$ has a locally identifiable transit target, then the corresponding orientation can be safely inferred. However, in general, deciding which target can serve as a valid transit node may require checking whether the resulting witness assignment creates additional inducing paths elsewhere in the graph.

We illustrate this obstruction with the following ADMG:

$$D = \{W \to X \to Y \to Z, \ X \to Z, \ Y \leftrightarrow Z\}, \qquad \mathcal{I} = \{\emptyset, \{X, Y\}\}.$$

Let $F$ be the intervention indicator corresponding to the pair of intervention targets in $\mathcal{I}$, so that $\mathrm{tar}(F) = \{X, Y\}$. The corresponding $\mathcal{I}$-MAG contains

$$W \to X \to Y \to Z, \qquad F \to X, \qquad F \to Y, \qquad F \to Z, \qquad X \to Z.$$

If we enumerate all valid $\mathcal{I}$-MAGs compatible with the learned partial graph, the edge $Y \rightarrow Z$ is invariant and hence identifiable. Nevertheless, FAST cannot recover this orientation using only the current local rules. The reason is that the non-target adjacency $F \rightarrow Z$ may locally be explained through either target $X$ or target $Y$, since both are candidate parents of $Z$. However, choosing $X$ as the transit node would create an inducing path from $W$ to $Z$, which would force $W$ and $Z$ to be adjacent in the $\mathcal{I}$-MAG. Since $W$ and $Z$ are not adjacent, $X$ cannot be a valid transit node; therefore $Y$ must be the transit node, which in turn compels $Y \rightarrow Z$.

This example shows that the missing information is not another purely local FCI-style implication, but rather a global realizability constraint: one must rule out transit assignments that create forbidden inducing paths between non-adjacent nodes. One could extend Rule 9 to cover this specific example, but similar obstructions can occur at larger graph distances. Thus, covering all such cases appears to require a recursive or global search over compatible witness assignments, which is precisely the role of the $\mathcal{I}$-MAG realizability oracle used by LIST. For this reason, FAST should be viewed as an efficient local refinement of $\mathcal{I}$-FCI , while LIST provides the theoretically complete but computationally more expensive completion procedure.

## G.2. Why Not FAST+LIST

A natural idea is to first apply FAST and then run LIST on the resulting partially oriented graph. Such a hybrid could be empirically attractive, since FAST may orient some marks before the more expensive enumeration step. However, we do not present FAST+LIST as a principled sound-and-complete algorithm because existing MAG-listing procedures assume that the input graph is PAG- or PMG-like (partial mixed graphs, see Zhang (2008)) and satisfies specific structural conditions, such as chordality of the circle component. The output of $\mathcal{I}$-FCI -base is designed to satisfy these requirements, but the FAST-refined graph is not guaranteed to remain in this admissible input class.

The issue is that FAST can orient circle–circle edges. Orienting such an edge effectively removes it from the circle component, and chordality is not preserved under edge removal: an edge that served as a chord may be removed, leaving a non-chordal cycle. Thus, applying MAG listing directly to the FAST output may fail the input conditions required by the current MAG-listing framework.

Consider the following ADMG:

$$D = \{W \rightarrow X \rightarrow Y, \ X \rightarrow Z, \ W \rightarrow Z, \ X \leftrightarrow W, \ W \leftrightarrow Z\}, \qquad \mathcal{I} = \{\emptyset, \{W, Y\}\}.$$

For the $\mathcal{I}$-FCI -base output $P_0$, the induced subgraph on $\{W, X, Y, Z\}$ is a circle component compatible with MAG listing. However, FAST orients $W \rightarrow Z$ by Rule 9, since the inducing path from $F$ to the non-target $Z$ must go through the target $W$. After this orientation, the resulting partially oriented graph need not satisfy the chordality and PMG-like assumptions required by the MAG-listing subroutine. Consequently, the procedure "run LIST on FAST$(P_0)$" is not currently covered by the correctness guarantees of existing MAG-listing algorithms.

To make a FAST+LIST hybrid rigorous, one would need either a MAG-listing algorithm that accepts more general partially oriented mixed graphs, or a provably safe way to incorporate the additional FAST orientations as constraints while preserving the structural invariants required by current MAG-listing methods. We leave this as an interesting direction for future work.

