# OpenReview forum: "Towards Completeness in Causal Discovery from Soft Interventions with Known Targets"
_ICML.cc/2026/Conference — ICML 2026 regular_

### Official Review · Reviewer_GYwi · 2026-02-19

**Soundness:** 3
**Presentation:** 3
**Significance:** 2
**Originality:** 3
**Overall Recommendation:** 4
**Confidence:** 4

**Summary:**

This paper studies the problem of interventional causal discovery with known targets in the presence of latent confounding. The authors show that the original algorithm I-FCI in Kocaoglu et al. (2019), which basically used the existing FCI orientation rules, is sound but not complete, i.e., it may leave circle endpoints in the output PAG that are in fact identifiable across all consistent intervention augmented MAGs. To resolve this, the concept of transit nodes is proposed to characterize the specific subclass of MAGs as "I-MAGs". Then, two methods are proposed: one is a theoretically complete enumeration-based method using the existing MAG listing algorithm, and another is the derivation of three new local orientation rules that are more efficient, but still incomplete.

**Compliance With Llm Reviewing Policy:**

Affirmed.

**Final Justification:**

I thank the authors for the response. I keep my positive score.

The position to background knowledge and scope wrt. unknown targets parts are explained clearly.

The reason for not scoring any higher is mainly due to the incremental (at least to me) technical contribution:
- The enumeration-based approach is basically built upon the existing works on MAG listing (of course I understand it's not directly reusing all the existing conclusions, and I appreciate the importance of the "realizability oracle" part. I just don't find this part highly technically challenging, as compared to MAG listing itself).
- As for the local rules' incompleteness, thanks for the new example. It would be beneficial to put them and more discussions into the current manuscript, to help readers get more sense on why they are incomplete, to what extent they are incomplete, and where the challenges lie. I look forward to seeing authors develop complete rules in their future work.

**Key Questions For Authors:**

see "weaknesses".

**Limitations:**

see "weaknesses".

**Strengths And Weaknesses:**

## Strengths:

1. The motivation for a complete algorithm for interventional causal discovery is clear and important.
2. The reasons for why the original FCI orientation rules are now incomplete are articulated in a clear way: adjacency patterns involving F-nodes encode structural constraints on inducing paths through intervention targets.
3. The concept of transit nodes provides a structured way to determine which MAGs are valid I-MAGs.
4. The running examples are clean and useful for understanding.

## Weaknesses:

1. **My major concern is that the scope studied seems too narrow, and the contribution appears marginal (both theoretically and empirically):**
   - For scope:
      - This work focuses on the setting with known targets, controlled soft interventions, and interventions only applied to observed variables.
      - Though the theoretical results in this work under this setting appears correct to me, this setting itself is too narrow.
      - In practice, cases are commonly that targets are unknown, or interventions applied to latent variables.
      - It would be valuable to discuss how the results in this work can imply or generalize to broader settings: e.g., whether the original I-FCI is complete in the unknown or latent targets settings.
   - For contribution:
      - Even under this narrow scope, the solutions are still not finished.
      - For the enumeration based solution, it depends on the existing prior work on MAG listing and does not add anything nontrivially new to that. Except for the worst-case super-exponential complexity, the empirical worse performance of this solution (as shown in Table 1) is also concerning.
      - For the local orientation rule based solution, these rules are still not complete. Though I acknowledge that completeness rules are challenging to construct (as seen in the development of original FCI), the authors should clearly discuss more on the reasons why the current rules are incomplete, and to what extent it is away from completeness. It is briefly mentioned in Figure 2 about the necessity of global conditions, and this part can be more in detail.

2. **The positioning relative to background knowledge literature needs more elaboration:**
   - This work relates closely to the works of FCI with background knowledge.
   - While this connection is mentioned, it can be elaborated more in detail. For example, in what sense are F-nodes more/less expressive/challenging than general local background knowledge? When the targets are unknown, can the only knowledge of "F nodes are root nodes" be completely characterized by existing works? Can the new rules be re-derived as special cases of extended BK rules?
   - Addressing these issues could perhaps strengthen the theoretical contribution of this work in a larger scope.

---

> ### Author Rebuttal · Authors · 2026-03-30
>
> We sincerely thank the reviewer for the careful reading and comprehensive feedback. Below we respond to each concern in detail.
>
> On narrow scope:
>
> We agree our formal results focus on a specific setting, and we will clarify this scope and its motivation.
>
> Regarding broader regimes, when targets are unknown, the setting is fundamentally different because the extra structure we exploit is not available. In that case, [1] provides a sound-and-complete causal discovery algorithm $\psi$-FCI. Therefore this problem is addressed. However, $\psi$-FCI has only 1 non-FCI rule that is orienting edges outwards from $F$ nodes, i.e. nothing more can be learned. In contrast, our proposed FAST has 3 additional rules but is not complete, showing the difficulty of our setting.
>
> For latent intervention targets, our current framework does not directly cover interventions on unobserved variables. This can be viewed as unknown target interventions, or equivalently, a soft intervention on the closest downstream observable nodes. Although this work focuses on controlled-experiment setting, it can be extended to uncontrolled setting if we define the targets based on their mechanisms as defined in [1].
>
> On contributions:
>
> We respectfully disagree that the enumeration-based approach is only a direct reuse of MAG listing. MAG listing enumerates candidate MAG completions. Our contribution is the I-MAG realizability characterization via transit nodes and the associated realizability-checking procedure, which filters candidate MAGs to the specific subclass consistent with valid I-MAGs. This realizability oracle is what enables theoretical completeness of I-MEC rather than observational MEC.
>
> On empirical performance, LIST is designed to be theoretically complete with a correctly learned skeleton, but can be more brittle under finite-sample CI errors because it enforces global consistency constraints. LIST may fail to return a completion when the initial learned skeleton is inconsistent with any realizable I-MAG while FAST only checks local structures and thus performs slightly better on average. If we input the right skeleton, LIST is consistently better than FAST.
>
> On incomplete local rules:
>
> We agree and will strengthen the discussion. The key obstruction is that some compelled orientations depend on global compatibility of witness assignments across multiple $(F,Y)$ pairs. Consider an example ADMG $D = \\{ W\to X\to Y\to Z, X\to Z, Y\leftrightarrow Z\\}$ with targets $\\{ \emptyset, \\{X, Y\\}\\}$. The I-MAG is $\\{ W\to X\to Y\to Z, F\to X\to Z, F\to Y, F\to Z\\}$. If we enumerate all valid I-MAGs, the edge $Y\to Z$ is invariant but it cannot be recovered by FAST. Notice that $Z$ has two possible parents $X, Y$ to be the transit node. If $X$ is the transit node, it will create an inducing path from $W$ to $Z$ making $W, Z$ adjacent in the I-MAG. However, since they are not adjacent, $Y$ has to be the transit node and we can orient $Y\to Z$. We can modify the current Rule 9 to include this case but obviously, there can be long distance effect that forbids some nodes to be the transit nodes which in turn orients more edges. To include all these structures, our conjecture is that one has to design a global recursive algorithm.
>
> On the positioning of this paper:
>
> We agree this can be improved and will expand the comparison. BK in the literature usually means causal knowledge directly input to the graphs. For example, local BK is defined as all the edge marks incident to a set of nodes ([2], [3]). Specifically, in the observational discovery setting, [3] has shown that with local BK, the FCI rules with modifications are sound and complete. However, in our setting, it is clear that FCI rules are incomplete with the local BK of $F$ nodes being root nodes. General BK can be any edge mark in the partially oriented graph. When targets are unknown, the primary guaranteed constraint is that $F$-nodes are exogenous root nodes, which closely resembles local BK of incident edge marks. Again, [1] has shown that $\psi$-FCI is complete. Therefore, the setting of unknown targets can be viewed as equivalent to the local BK of $F$ nodes being root nodes. In contrast, with known targets, the semantics include also the known child set $tar(F)$ and the fact that $F$-to-non-target adjacency arises only through inducing paths mediated by targets. These are higher-order constraints about inducing paths and target-mediated structure, and are not equivalent to simply fixing a subset of edge marks.
>
> We will add the discussions to the revision. We believe the proposed revisions will meaningfully improve clarity and impact, and we welcome any additional questions or suggestions.
>
> [1] "Causal discovery from soft interventions with unknown targets: Characterization and learning."
>
> [2] "New rules for causal identification with background knowledge."
>
> [3] "Sound and complete causal identification with latent variables given local background knowledge."

---

> > ### Author Rebuttal · Reviewer_GYwi · 2026-04-02
> >
> > I thank the authors for the response. I keep my positive score.
> >
> > The position to background knowledge and scope wrt. unknown targets parts are explained clearly.
> >
> > The reason for not scoring any higher is mainly due to the incremental (at least to me) technical contribution:
> > - The enumeration-based approach is basically built upon the existing works on MAG listing (of course I understand it's not directly reusing all the existing conclusions, and I appreciate the importance of the "realizability oracle" part. I just don't find this part highly technically challenging, as compared to MAG listing itself).
> > - As for the local rules' incompleteness, thanks for the new example. It would be beneficial to put them and more discussions into the current manuscript, to help readers get more sense on why they are incomplete, to what extent they are incomplete, and where the challenges lie. I look forward to seeing authors develop complete rules in their future work.

---

### Official Review · Reviewer_3xt3 · 2026-03-06

**Soundness:** 2
**Presentation:** 3
**Significance:** 2
**Originality:** 3
**Overall Recommendation:** 4
**Confidence:** 4

**Summary:**

This paper studies how to use the controlled soft intervention data of known intervention targets for causal structure learning in the presence of latent confounders.
The authors propose two methods for this problem.
First, based on MAG enumeration, they propose a sound and theoretically complete method; however, this method may be computationally expensive. Furthermore, they propose three new orientation rules as an extension of I-FCI (Kocaoglu et al., 2019), which are not complete but are computationally efficient.

**Compliance With Llm Reviewing Policy:**

Affirmed.

**Final Justification:**

This work is motivated by the goal of developing a complete variant of I-FCI, following Kocaoglu et al. (2019), who introduced two methods. The first, based on MAG enumeration, is sound and theoretically complete; however, it is computationally expensive. The second, based on three new orientation rules, is computationally efficient but incomplete.

The authors have addressed my questions, and I have increased my score from 3 to 4. However, some limitations still remain. In particular, FAST’s gap to completeness remains unclear, and more broadly, there is still no method in their setting that is both complete and computationally efficient, comparable to FCI.

**Key Questions For Authors:**

- In Figure 1(e), the statement ``$W$ is not a target but adjacent to $F$. This could happen only when there is an inducing path from $F$ to $W$ via a target'' is not trivial. Could the authors clarify this statement? In particular, does this claim hold relative to the graph shown in Figure 1(b)?

- Why is the assumption of ''assuming
exhaustive I-MAG completion enumeration'' needed in Theorem 5.8? Since the method in Wang et al. (2025) does not provide a way to enumerate all valid I-MAGs relative to the $P_0$?

- The experiment considered a very small number of nodes (5). If the number of nodes were larger, such as 50 or 100, would FAST exhibit significantly higher efficiency while maintaining accuracy comparable to LIST?

- In the experimental part, the maximum size of intervention target is 2 and the number of domains is 3, Could the authors consider more general settings?

- Could combining FAST and LIST (i.e., FAST+LIST) achieve both higher efficiency and completeness?

- The proposed rules are designed only for non-target nodes adjacent to $F$ nodes. Is there a possible extension for the orientation of edges between variables without $F$ nodes?

- In the Algorithm 1, depth-first search is used. Have the authors considered other search strategies?

**Limitations:**

See the Weaknesses section.

**Strengths And Weaknesses:**

# Strengths:

- The overall technical level of the paper is rigorous, although I did not verify the proofs in the appendix.

- Based on the existing theoretical results of I-MAG and I-FCI, the authors clearly point out the incompleteness of the existing methods under the setting that soft intervention data of known intervention targets in the presence of latent confounders.

- The overall structure of the paper is clear. They propose three new orientation rules as an extension of I-FCI (Kocaoglu et al., 2019).

# Weaknesses:

- The proposed method is limited to soft interventions with known intervention targets. It cannot be generalized to settings with general interventions [1] or to cases where the intervention targets are unknown [2,3].

[1] Characterizing and Learning Equivalence Classes of Causal DAGs under Interventions. ICML 2018.

[2] Causal Discovery from Soft Interventions with Unknown Targets: Characterization and Learning. NeurIPS 2020.

[3] Causal discovery from observational and interventional data across multiple environments.
NeurIPS 2023.

- The proposed LIST method relies on the MAG-listing method in Wang et al. (2025), which can enumerate all valid MAGs for a PAG learned from observational data. However, it is unclear whether this method can enumerate all valid I-MAGs for a PAG with $F$ nodes obtained from I-FCI. The paper does not provide theoretical results addressing this issue. There is a gap left here.

- The scale of the experiment is limited. The generated graphs have very small dimensionality (only 5 nodes). Moreover, the paper does not include experiments on real-world data, making it difficult to demonstrate the practical significance of the proposed method.

- Although the paper provides some examples with graphs, some key technical definitions (e.g., transit sets, candidate transit nodes, and transit selection sets) lack illustrative examples and are not intuitive enough, which makes the paper difficult to follow.

---

> ### Author Rebuttal · Authors · 2026-03-30
>
> We sincerely thank the reviewer for the careful reading and comprehensive feedback. Below we respond to each concern in detail.
>
> On limitation of scope:
>
> We agree our current theoretical development focuses on controlled soft interventions with known targets, because the augmented-graph semantics are particularly informative in this setting, and prior work already establishes completeness in some neighboring regimes.
>
> [1] considers general interventions, but does not handle latent confounders.
>
> [2] studies soft interventions with unknown targets and provides a sound and complete causal discovery algorithm $\psi$-FCI in that setting. Specifically, $\psi$-FCI has 1 rule in addition to 7 FCI rules that is orienting edges outwards from $F$ nodes. Our work instead targets the complementary regime of known targets, where we show the FCI-style closure is incomplete and we add additional orientation principles. Although the unknown target setting is more general, this paper shows there is extra information by knowing the intervention targets and how to use it in discovery.
>
> [3] proposes S-FCI for mixtures of observational and interventional data across multiple environments and proves soundness, including a targeted orientation rule (Rule 9') in a special known-target case with a single target. Similarly, it does not extend the orientation principle to I-FCI when the targets are known so this gap still exists. Our Rule 9 can be viewed as pushing the same “identifiable inducing path via targets” idea beyond the single-target special case and we can add a short discussion positioning our rules relative to Rule 9’. Besides, it is straightforward to integrate our extra orientation rules in FAST to improve S-FCI.
>
> On enumeration of valid I-MAGs:
>
> This is a fair request for clarification, and we will make it explicit in the revision. Every valid I-MAG is, by definition, a MAG over $V\cup F$. Hence, if we enumerate all MAG completions consistent with $P_0$, we cannot miss an I-MAG due to the listing step alone. While $P_0$ is not an observable PAG in the standard sense, in our setting it is equivalent to running PAG reasoning with local background knowledge around $F$, so it satisfies the graphical properties required by the MAG listing input contract. Alg. 2 therefore lists all MAG completions consistent with $P_0$, then uses Alg. 1 as an I-MAG realizability oracle to filter the subset that are valid I-MAGs, and finally intersects marks across those valid I-MAGs. We also included this in Thm. 5.8 as an assumption.
>
> On lacking illustrative examples:
>
> We appreciate this feedback and agree that the concepts would benefit from a more concrete illustration. In the revision we will add a step-by-step running example for each concept.
>
> On scaling to large graphs and realistic setting:
>
> We run experiments on $n=100$ nodes and report the performances. Please refer to our response to MmD9 and 5uTv.
>
> On Fig. 1:
>
> This follows from the definition of I-MAG as a latent projection of the augmented graph: adjacency $F\to W$ in the I-MAG implies there is an inducing path between $F$ and $W$ in the augmented graph. Since $F$ has outgoing edges only into targets, any such inducing path must begin by going from $F$ to some target $X \in tar(F)$. This is why $F$ can become adjacent to non-targets in the I-MAG, even though it only points to targets in the augmented graph. Accordingly in Fig. 1(b), there is an inducing path from $F$ to $W$ through $X$.
>
> On more general experimental settings:
>
> Here we report results on 2 domains and different maximum target size with 10 nodes. Each setting takes 50 runs. We will add more settings in the revision due to character limit here.
>
> |METHOD|Max|PG SHD|Runtime|
> |------ |-:|-----------:|------------:|
> |I-FCI|1|24.16±4.85|0.060±0.003|
> |FAST|1|24.14±4.85|0.061±0.004|
> |LIST|1|24.15±3.30|2.543±0.879|
> |I-FCI|2|24.08±5.19|0.060±0.002|
> |FAST|2|23.80±5.24|0.061±0.003|
> |LIST|2|22.75±3.82|2.469±0.919|
> |I-FCI|3|23.58±5.00|0.060±0.003|
> |FAST|3|23.24±5.12|0.061±0.003|
> |LIST|3|22.36±4.27|2.435±0.999|
>
>
> On FAST+LIST:
>
> Good suggestion. This should improve the efficiency of pure LIST empirically.
>
> On orientation rules:
>
> Our current additional rules specifically exploit $F$-adjacency because that is where interventions provide extra constraints beyond standard FCI reasoning. Edges not connected to $F$-adjacency are still oriented by the standard FCI closure. Beyond that, completeness is guaranteed by LIST. Our conjecture is that structure outside $F$-adjacency would affect the orientation inside while the rules will not orient outside nodes.
>
> On search algorithms:
>
> DFS is an implementation choice for exploring a bounded-branching combinatorial space with early pruning. Other backtracking strategies could be used, but do not change correctness.
>
> We will add the discussion to the revision. We believe the proposed revisions will meaningfully improve clarity and impact, and we welcome any additional questions or suggestions.

---

> > ### Author Rebuttal · Reviewer_3xt3 · 2026-04-02
> >
> > Thank you for your response.
> >
> > I am still curious whether FAST+LIST also enjoys soundness and completeness; intuitively, this seems plausible. If so, it would be helpful to understand why the authors did not include this seemingly straightforward variant. I would also be interested in its computational efficiency and PG-SHD compared with FAST, as this seems important for understanding FAST’s gap in completeness.
> >
> > I will increase my score from 3 to 4.

---

> > > ### Author Response · Authors · 2026-04-02
> > >
> > > Thank you for the follow-up questions and for taking another look at our work.
> > >
> > > We agree that a hybrid FAST+LIST is intuitively attractive. However, there is a technical gap that prevented us from including it as a principled variant in the current submission.
> > >
> > > Concretely, our LIST procedure relies on existing MAG-listing algorithms that assume the input graph is PAG or PMG-like and satisfies specific graphical constraints (e.g., the circle component must be chordal, along with other structural conditions required by the local-transformation framework [1], [2]). While the I-FCI-base output $P_0$ is designed to satisfy these constraints, the FAST-refined graph is not guaranteed to remain in the admissible input class for MAG listing. In particular, FAST orients some circle–circle edges. This can remove edges from the circle component, and chordality is not preserved under edge removal since an edge that served as a chord may be removed, leaving a non-chordal cycle.
> > >
> > > Consider this exmaple ADMG $D = \\{ W \to X \to Y, X \to Z, W \to Z, X\leftrightarrow W, W\leftrightarrow Z\\}$ with targets $\\{\emptyset, \\{W, Y\\} \\}$. The induced subgraph on $W, X, Y, Z$ in $P_0$ is a circle component compatible for MAG listing. However, FAST will orient $W \to Z$ breaking the chordality according to Rule 9 since the inducing path to Z have to go through W. As a result, the FAST output may fail the MAG-listing input checks, so “run LIST on FAST($P_0$)” is not currently covered by existing correctness guarantees.
> > >
> > > To make FAST+LIST work rigorously, one would need either a MAG-listing procedure that can enumerate completions from a more general partially oriented mixed graph (beyond the current PMG assumptions), or a provably safe way to incorporate FAST’s extra orientations as constraints while preserving the structural invariants required by current MAG listing.
> > >
> > > We will add a brief example and remark clarifying this issue in the revision. We also agree it is an interesting direction for future work, since it could potentially yield a practically faster completion procedure while retaining completeness.
> > >
> > > If the above addresses your concern, we would appreciate it if you could update your recommendation accordingly.
> > >
> > > References:
> > >
> > > [1] Wang, Tian-Zuo, Wen-Bo Du, and Zhi-Hua Zhou. "An efficient maximal ancestral graph listing algorithm." In Forty-first International Conference on Machine Learning. 2024.
> > >
> > > [2] Wang, Tian-Zuo, Wen-Bo Du, and Zhi-Hua Zhou. "Polynomial-delay mag listing with novel locally complete orientation rules." In Forty-second International Conference on Machine Learning. 2025.

---

### Official Review · Reviewer_5uTv · 2026-03-13

**Soundness:** 3
**Presentation:** 3
**Significance:** 4
**Originality:** 4
**Overall Recommendation:** 5
**Confidence:** 3

**Summary:**

This paper studies  causal discovery from soft interventions in the presence of latent confounding. To formalize this setting, the paper proposes  an enumeration-based completion procedure for multi-domain interventional data.

**Compliance With Llm Reviewing Policy:**

Affirmed.

**Final Justification:**

I have no further concern on this work. The authors should consider incorporating these newly added results into the main paper, as they help clarify the method’s effectiveness and practical value.

**Key Questions For Authors:**

N/A

**Limitations:**

yes

**Strengths And Weaknesses:**

Strengths:

1. Learning causal structure from observational data is a central problem in causal discovery.

2. The work extends ideas from classical causal discovery algorithms (e.g., FCI and PAG theory) to the interventional setting, which is technically nontrivial.

3. Experiments on synthetic data show that the fast rule-based refinement and enumeration-based completion procedure can recover more edge marks than I-FCI.

4. The theoretical analysis and presentation of the paper are solid, and the code is also available.

Weaknesses:

It is still unclear how the proposed method can be applied in real-world settings. Although causal discovery with latent variables is an important problem, the scalability of existing algorithms on large datasets remains a challenge.

The experimental section is somewhat limited. Although it demonstrates the effectiveness of FAST, it does not evaluate performance across different sample sizes or numbers of variables.

---

> ### Author Rebuttal · Authors · 2026-03-30
>
> We thank the reviewer for the comprehensive and thoughtful feedback. We address the concerns and questions point-by-point below.
>
> On lacking real-world evaluations:
>
> We agree that stronger benchmarks would be valuable. To address this, we test I-FCI, FAST, and LIST on the Sachs dataset [1]. In each run, we pick two domains randomly. We treat the nodes whose mechanisms change across experiments as the targets. In some cases, the true perturbed node is not among the observable nodes, we then use the closest downstream observable nodes as the targets. We compare the output partial graphs to the reference graph derived from the literature. We use the partial graph SHD (count of mismatching edge marks) as the metric. Lower values mean more edge marks are aligned. The outcomes for I-FCI, FAST, and LIST are $20.00 \pm 0.55, 19.80 \pm 0.45$, and $19.20 \pm 0.84$ respectively. The runtimes are $0.0897 \pm 0.0065, 0.0994 \pm 0.0167$, and $8.1648 \pm 0.6423$ seconds respectively. Here the performance does not differ much with FAST and LIST being slightly better than I-FCI while LIST is much slower. This can arise from weak latent-confounding effects in the dataset so that our orientation principle does not provide extra information. Note that the reference graph is not a ground truth. We will add the experiment details and the learned graphs in the revision. We will also add experiments with semi-realistic simulators [2] so that we have access to the ground truth graphs.
>
> On limited experimental settings:
>
> We add an experiment on 100 observable nodes. For each run, we randomly generate a sparse ADMG and pick 2 targets with maximum target size of 7. We run 50 times and record the mean and std. On partial graph SHD, I-FCI and FAST are $813.20 \pm 28.59$ and $810.70 \pm 28.63$ respectively. The runtime for I-FCI and FAST are $105.21 \pm 0.13$ and $105.23 \pm 0.17$ seconds respectively. While LIST takes more than 500 seconds for 1 run so we terminated LIST. We can see that the improvement of FAST based on I-FCI is small because the rules of FAST only trigger at some specific configurations which may not always appear in the graphs, especially for sparse graphs. We also notice that FAST and I-FCI take almost the same amount of time, showing the efficiency of FAST.
>
> We will add the experiments with more settings in the revision.
>
> We also compare the performances with $n=10$ observable nodes with different sample sizes.
>
> |     N | FAST Skel-SHD |  FAST PG-SHD | FAST Runtime (s) | I-FCI Skel-SHD | I-FCI PG-SHD | I-FCI Runtime (s) | LIST Skel-SHD |  LIST PG-SHD | LIST Runtime (s) |
> | ----: | ------------: | -----------: | ---------------: | -------------: | -----------: | ----------------: | ------------: | -----------: | ---------------: |
> |  2500 |  14.16 ± 3.30 | 23.52 ± 4.09 | 0.08 ± 0.01 | 14.16 ± 3.30 | 23.72 ± 4.11 | 0.08 ± 0.01 |  14.16 ± 3.30 | 23.52 ± 4.09 |    22.59 ± 28.76 |
> |  5000 | 13.04 ± 3.53 | 23.58 ± 5.61 | 0.14 ± 0.01 | 13.04 ± 3.53 | 23.82 ± 5.63 | 0.14 ± 0.01 | 13.04 ± 3.53 | 23.66 ± 5.33 | 38.38 ± 40.26 |
> | 10000 | 13.30 ± 3.19 | 23.74 ± 5.05 | 0.27 ± 0.01 | 13.30 ± 3.19 | 24.22 ± 4.82 | 0.26 ± 0.01 | 13.30 ± 3.19 | 23.78 ± 4.64 | 41.67 ± 44.93 |
> | 25000 | 12.28 ± 3.15 | 24.20 ± 4.74 | 0.64 ± 0.04 | 12.28 ± 3.15 | 24.70 ± 4.61 | 0.64 ± 0.04 | 12.28 ± 3.15 | 24.44 ± 4.68 | 68.92 ± 40.93 |
> | 50000 | 11.94 ± 3.09 | 24.76 ± 5.34 | 1.25 ± 0.08 | 11.94 ± 3.09 | 25.08 ± 5.26 | 1.25 ± 0.08 | 11.94 ± 3.09 | 24.38 ± 4.83 | 69.65 ± 41.26 |
>
> We observe that skeleton SHD improves monotonically with sample size, which matches the intuition that more data improves CI testing and hence adjacency recovery. The non-monotonicity appears only in PG-SHD, which penalizes edge marks and compares  against a single reference MAG. With larger sample size $N$, the algorithms typically commit to more orientations. These additional orientation commitments can increase mark-based error even when the underlying skeleton is more accurate, especially when some orientations are not identifiable from the available interventional information, or CI-test errors propagate through the orientation rules, or the evaluation compares a partial output to a single MAG reference rather than the correct equivalence-class object.
>
> To make this transparent, we will report skeleton-only metrics separately (e.g., skeleton SHD) alongside mark metrics, and we will clarify the evaluation target used for PG-SHD.
>
> We believe these clarifications and additions will strengthen the manuscript, and we are happy to address any further questions.
>
> References:
>
> [1] Sachs, Karen, Omar Perez, Dana Pe'er, Douglas A. Lauffenburger, and Garry P. Nolan. "Causal protein-signaling networks derived from multiparameter single-cell data." Science 308, no. 5721 (2005): 523-529.
>
> [2] Dibaeinia, Payam, and Saurabh Sinha. "SERGIO: a single-cell expression simulator guided by gene regulatory networks." Cell systems 11, no. 3 (2020): 252-271.

---

> > ### Author Rebuttal · Reviewer_5uTv · 2026-04-03
> >
> > I appreciate the authors’ responses and clarifications. I will increase my confidence from 1 to 3.

---

### Official Review · Reviewer_MmD9 · 2026-03-13

**Soundness:** 4
**Presentation:** 3
**Significance:** 3
**Originality:** 3
**Overall Recommendation:** 5
**Confidence:** 4

**Summary:**

The paper proposes a method for using soft interventions to refine orientation in I-PAG, obtained using the I-FCI algorithm.
First, the authors demonstrate that I-FCI rules are not complete, i.e., orientations obtained due to invariance are not present in the final I-PAG. They then proceed to provide an enumeration procedure for MAGs consistent with the output I-FCI and identify valid I-MAGs. Ultimately, they update three of the I-FCI rules to yield a tighter equivalence class. These rules enforce consistencies derived from the presence of inducing paths and transit sets in the augmented graph.

**Compliance With Llm Reviewing Policy:**

Affirmed.

**Final Justification:**

The rebuttal addressed my concerns.

**Key Questions For Authors:**

1. Can the results in the paper be extended to hard interventions? How? If not, why?
2. Do you think that this method could be easily extended to unknown target interventions, such as in [2]?
3- Can define more clearly the notion of multi domain?

[2] Jaber, Amin, Murat Kocaoglu, Karthikeyan Shanmugam, and Elias Bareinboim. “Causal discovery from soft interventions with unknown targets: Characterization and learning.” Advances in neural information processing systems 33 (2020)

**Limitations:**

Yes

**Strengths And Weaknesses:**

Soundness:

* The paper is well written and clear. All theoretical results are accompanied by an intuitive explanation of their meaning and use. Additional graphical examples are provided to further clarify the method.
* The paper delves into an important topic, providing significant contributions. Identifying a tighter equivalence class in the presence of unmeasured confounders is a step toward a more practical use of causal discovery methods.

Weaknesses:

* I think the paper would benefit from a deeper characterisation of the notion of “multi-domain”. Does this relate to the notion of environments in [1]?
* The impact of the paper would benefit from additional results over more realistic datasets.

Minor:

* line 076 col 2: BK is not defined previously
* line 232 col 2: ADMG… -> ADMG.
* In the notation section, defining S as a subset of V would improve clarity.

[1] Li, A., Jaber, A., and Bareinboim, E. Causal discovery from observational and interventional data across multiple environments. Advances in Neural Information Processing Systems 36 (2023)

---

> ### Author Rebuttal · Authors · 2026-03-30
>
> We thank the reviewer for the comprehensive and thoughtful feedback. We address the concerns and questions point-by-point below.
>
> On multi-domain definition:
>
> In our paper, each domain corresponds to a single soft-interventional regime, i.e., one interventional distribution $P_I$ induced by a known intervention target set $I$. We will revise the manuscript to make this definition explicit early. This is related to—but distinct from [1], which considers a broader multi-domain setting (e.g., multiple distributions per domain or unknown targets). Although the setting is broader in [1], the gap of learning from soft interventions with known targets was not solved. Our focus is the known-target soft-intervention setting and we explore the completeness guarantees for this setting, which complements existing work.
>
> On lacking evaluation on real-world datasets:
>
> We agree that stronger benchmarks would be valuable. To address this, we test I-FCI, FAST, and LIST on the Sachs dataset [4]. In each run, we pick two domains randomly. We treat the nodes whose mechanisms change across experiments as the targets. In some cases, the true perturbed node is not among the observable nodes, we then use the closest downstream observable nodes as the targets. We compare the output partial graphs to the reference graph derived from the literature. We use the partial graph SHD (count of mismatching edge marks) as the metric. Lower values mean more edge marks are aligned. The outcomes for I-FCI, FAST, and LIST are $20.00 \pm 0.55, 19.80 \pm 0.45$, and $19.20 \pm 0.84$ respectively. The runtimes are $0.0897 \pm 0.0065, 0.0994 \pm 0.0167$, and $8.1648 \pm 0.6423$ seconds respectively. Performances do not differ much. FAST and LIST are slightly better than I-FCI while LIST is much slower. This can arise from weak latent-confounding effects in the dataset so that our orientation principle does not provide extra information. Note that the reference graph is not a ground truth. We will add the experiment details and the learned graphs in the revision. We will also add experiments with semi-realistic simulators [3] so that we have access to the ground truth graphs.
>
> Issues on typos and definitions:
>
> Thank you for pointing out the typos and undefined terms (e.g., BK). We will correct these in the revision.
>
> On extension to hard interventions:
>
> Extending to hard interventions is non-trivial because the learning objective changes: hard interventions remove incoming edges into targets, yielding different separation relations across regimes and typically requiring a multi-graph objective. Our approach leverages soft-intervention structure where the underlying causal graph remains fixed and regime effects are captured via F-nodes. Furthermore, the inducing-path constraints we exploit do not carry over directly under edge-deletion interventions. To witness, if a node is hard-intervened, any incoming edges into the node are removed and thereby breaking any inducing path through that node. We will add a brief discussion clarifying this distinction.
>
> Question on extension to unknown targets:
>
> Thanks for this question. Unknown-target soft interventions have been studied in [2] and $\psi$-FCI provides both soundness and completeness in that setting. Specifically, in addition to the 7 FCI rules, psi-FCI only needs one extra rule that orients edges outwards from $F$ while FAST proposed in this paper has 3 extra rules but is not complete yet. Right now, we have to rely on a MAG listing approach for completeness. In this sense, the setting of unknown targets is easier to solve. We will add a discussion on this to clarify our contribution in the revision.
>
> We believe these clarifications and additions will strengthen the manuscript, and we are happy to address any further questions.
>
> References:
>
> [1] Li, Adam, Amin Jaber, and Elias Bareinboim. "Causal discovery from observational and interventional data across multiple environments." Advances in Neural Information Processing Systems 36 (2023): 16942-16956.
>
> [2] Jaber, Amin, Murat Kocaoglu, Karthikeyan Shanmugam, and Elias Bareinboim. "Causal discovery from soft interventions with unknown targets: Characterization and learning." Advances in neural information processing systems 33 (2020): 9551-9561.
>
> [3] Dibaeinia, Payam, and Saurabh Sinha. "SERGIO: a single-cell expression simulator guided by gene regulatory networks." Cell systems 11, no. 3 (2020): 252-271.
>
> [4] Sachs, Karen, Omar Perez, Dana Pe'er, Douglas A. Lauffenburger, and Garry P. Nolan. "Causal protein-signaling networks derived from multiparameter single-cell data." Science 308, no. 5721 (2005): 523-529.

---

> > ### Author Rebuttal · Reviewer_MmD9 · 2026-04-02
> >
> > I thank the authors for their response. I believe the modification they promised to do will improve the clarity of the paper.

---

### Decision · Program_Chairs · 2026-04-30

**Decision:**

Accept (regular)

**Comment:**

The reviewers agreed that this paper addresses a meaningful problem in causal discovery from soft interventions with known targets under latent confounding. They found the main technical contribution clear: the paper shows that I-FCI is sound but not complete in this setting, and proposes both a theoretically complete completion procedure and a more efficient rule-based refinement. The final assessments were positive overall, and multiple reviewers indicated that the rebuttal addressed their main concerns.

Overall, I recommend acceptance. The main concerns raised in discussion were about the narrow scope of the setting, the practical scalability of the enumeration-based method, the remaining gap between FAST and completeness, and the need for stronger empirical validation and clearer positioning relative to related work. After reading the paper and the author responses carefully, I find that the main concerns were addressed sufficiently to support publication, although some limitations remain, particularly regarding the incompleteness of the efficient rule-based method and the absence of a method in this setting that is both complete and computationally efficient. The paper nevertheless makes a technically solid contribution in a challenging setting, and the final reviewer assessments support publication.